



# MPR 1.0: A stand-alone Multiscale Parameter Regionalization Tool for Improved Parameter Estimation of Land Surface Models

Robert Schweppe[1,2], Stephan Thober[1], Matthias Kelbling[1], Rohini Kumar[1], Sabine Attinger[1,2], and Luis Samaniego[1]

[1]Helmholtz-Centre for Environmental Research - UFZ, Permoserstraße 15, 04315 Leipzig, Germany
[2]Institute of Earth and Environmental Science, University of Potsdam - Karl-Liebknecht-Str. 24–25 14476, Potsdam-Golm, Germany

**Correspondence:** Stephan Thober (stephan.thober@ufz.de)

**Abstract.** Distributed environmental models such as land surface models (LSM) require model parameters in each spatial modelling unit (e.g. grid cell), thereby leading to a high-dimensional parameter space. One approach to decrease the dimensionality of parameter space in these models is to use regularization techniques. One such highly efficient technique is the Multiscale Parameter Regionalization (MPR) framework that translates high-resolution predictor variables (e.g., soil textural properties) into model parameters (e.g., porosity) via transfer functions (TFs) and upscaling operators that are suitable for every modeled process. This framework yields seamless model parameters at multiple scales and locations in an effective manner. However, integration of MPR into existing modeling workflows has been hindered thus far by hard-coded configurations and non-modular software designs. For these reasons, we redesigned MPR as a model-agnostic, stand-alone tool. It is a useful software for creating graphs of netCDF variables, wherein each node is a variable and the links consist of TFs and/or upscaling operators. In this study, we present and verify our tool against a previous version, which was implemented in the mesoscale hydrologic model mHM (www.ufz.de/mhm). By using this tool for the generation of continental-scale soil hydraulic parameters applicable to different models (Noah-MP and HTESSEL), we showcase its general functionality and flexibility. Further, using model parameters estimated by the MPR tool leads to significant changes in long-term estimates of evapotranspiration, as compared to their default parameterizations. For example, a change of up to 25% in long-term evapotranspiration flux is observed in Noah-MP and HTESSEL in the Mississippi River basin. We postulate that use of the stand-alone MPR tool will considerably increase the transparency and reproducibility of the parameter estimation process in distributed (environmental) models. It will also allow a rigorous uncertainty estimation related to the errors of the predictors (e.g., soil texture fields), transfer function and its parameters, and remapping (or upscaling) algorithms.





# 1 Introduction

Distributed environmental models simulate key fluxes and states of the atmosphere, land surface, and subsurface for a given spatial domain and time period (e.g., CLM (Andre et al., 2020), JULES (Best et al., 2011), or ORCIDEE (Krinner et al., 2005)). The underlying physical processes are simplified with parameterizations that are manifested as computer algorithms. Parameterizations are idealized representations of reality and as such there is inherent uncertainty in their formulation. They

require additional variables and model parameters in order to perform simulations. The latter could be constant, or it could be spatially and temporally variable over the simulation domain (i.e., the so-called distributed model parameters). Constant model parameters do not allow accurate characterization of environmental processes over a range of climatic regimes and geo-physical properties (Samaniego et al., 2010; Beck et al., 2016). The number of distributed model parameters tend to scale linearly with the number of spatio-temporal units, which is defined by the coordinates along each dimension of the

parameterized process. Model parameters are often fine-tuned to match observed fluxes and states of physical processes in various fields, such as hydrology (Duan et al., 1992; Gupta et al., 1998; Zink et al., 2017; Pagliero et al., 2019), Earth System Sciences (Sellers et al., 1989; Troy et al., 2008), and hydraulics (Shoarinezhad et al., 2020). Currently, there exists a plethora of methods for parameter estimation (Samaniego et al., 2017), which is the process of estimating a set of model parameters over the whole domain and their respective distribution functions. Such methods include classification linked through lookup

tables (for example, ECMWF, 2019; Andre et al., 2020), direct calibration (for example, Li et al., 2018; Arheimer et al., 2020), calibration and regionalization (for example, Carrera et al., 2005; Oudin et al., 2008; Samaniego et al., 2010; Rojas-Serna et al., 2016; Hundecha et al., 2016), and probabilistic methods (for example, González-García et al., 1998; Thiemann et al., 2001). Optimization issues related to parameter estimation through calibration tasks are often solved by the application of optimization algorithms (for example, Duan et al., 1993; Tolson and Shoemaker, 2007).

Distributed models have a parameter space with a very large dimensionality (Schaake, 2003). Even if high-performance computing (HPC) is available, it is not only numerically infeasible to estimate parameters for every individual grid cell, but also an ill-posed problem in lieu of limited availability of reference datasets (Kirchner, 2006). Samaniego et al. (2017), for example, indicated the presence of large differences in parameter estimation methods and the derived distributed parameter fields for many state-of-the-art global hydrological models (GHMs) and land surface models (LSMs). Regionalization techniques, such

as multiscale parameter regionalization (MPR) (Samaniego et al., 2010), provide an approach for reduction of dimensionality of parameter space through efficient use of regularization functions to estimate spatially explicit model parameters (Pokhrel and Gupta, 2010; Gupta et al., 2014).

The MPR framework operates on a two-step procedure. In a first step, it employs transfer functions (TFs) to translate high-resolution geo-physical properties into high-resolution model parameters. It can easily meet the requirements posed by

Van Looy et al. (2017) to couple information from different datasets (e.g., soil, vegetation, and topography) or establish time-varying parameters depending on changes in land use or climate (Vereecken et al., 2019). In the second step, high-resolution parameters are upscaled using upscaling operators to the spatial resolution and topology of the selected spatial units at which the model is to be applied. The resulting parameters are quasi-scale-independent and explicitly consider sub-grid variability



and predictor uncertainty if multiple sources are used (Samaniego et al., 2010; Kumar et al., 2013b; Rakovec et al., 2016;
Dembélé et al., 2020). Optimization approaches therefore should adjust the (globally applied) parameters of the TFs and the
parameters of the upscaling operator.

Currently, MPR is implemented as part of the source code of the mesoscale hydrologic model (mHM) (Samaniego et al.,
2010; Kumar et al., 2013b) and cannot be easily adapted for other models. Furthermore, this mHM-bound version is restricted
to rectangular grids and uses a hard-coded set of TFs for model parameters required by the mHM specifically. Mizukami et al.
(2017) proposed a flexible version of MPR (`MPR-flex`) to estimate parameters for the hydrological models VIC (Liang et al.,
1994; Hamman et al., 2018) and SAC (Burnash, 1995). This tool is also limited to a set of model-specific parameters, namely:
TFs and its two targeted models. In recent years, there have been numerous applications of MPR as a parameter estimation
technique for other models (for example, Samaniego et al., 2017; Mizukami et al., 2017; Imhoff et al., 2020); however, no
generic software currently exists.

More recently, Gou et al. (2021) implemented their own version of MPR on several parameters of the VIC model. In this
case, MPR allowed then to generate the first seamless (i.e., spatiotemporal continuous) reconstruction of long-term naturalized
runoff for China. In this study, they also found that the MPR technique was the best approach for regionalization among several
methods applied. It should be noted, however, that implementing MPR from scratch to any model is a time consuming and
cumbersome task because it demands large modifications of models' source code. In average, implementing MPR demands at
least one year of work of an expert scientific programmer. Hence, few groups in the world have attempted so far, as shown by
the references above.

In addition, existing applications are more targeted toward hydrologic applications; nevertheless, challenges persist with
regards to accurate estimation of the seamless fields of model parameters across a variety of spatial resolutions in different
compartments of Earth System Science models.

Therefore, a new `MPR` tool that provides a tailored framework for distributed parameter estimation is urgently needed (Van Looy
et al., 2017; Vereecken et al., 2019). With this aim in mind, we propose an MPR framework that can be used as a preprocessor
for both large-scale applications of land-surface models and global or regional hydrologic models. It needs to be a flexi-
ble, model-agnostic, light-weight, and high-performance tool with few external dependencies. Another key goal is to allow the
`MPR` tool to be embedded in optimization workflows such that TFs and remapping techniques can be easily modified (Van Looy
et al., 2017). Thus, the configuration overhead should be kept minimal. Although targeted towards and originating from the
LSM community, the development of MPR is aimed at supporting parameter estimation for distributed models in any scientific
field.

One key challenge is establishing a proper linkage between model parameters and suitable predictor variables (Clark et al.,
2016; Blöschl et al., 2019). Most currently-used TFs are derived from commonly observed or measured predictors and param-
eters (Van Looy et al., 2017). Mathematical frameworks (e.g., linear regression models, artificial neural networks, and random
forests) are applied to training datasets in order to develop functional relationships. However, TFs can also be inferred through
inverse methods. Emerging methods for the development of TFs do exist (Klotz et al., 2017; Feigl et al., 2020; Merz et al.,
2020). MPR provides the interface to link these tools to distributed environmental models.





Providing a library of remapping schemes is crucial, as the parameterizations in environmental models are not applied

on the scales at which they were derived. For example, the Richardson & Richards' equation (Richardson, 1922; Richards, 1931) describing unsaturated water flow through porous media at the representative elemental volume scale is often used at the landscape scale (say $10^2$ km). The inherent uncertainty of the physical parameters describing this phenomenon needs to be adequately considered by the choice of transfer functions and upscaling operators (Montzka et al., 2017; Vereecken et al., 2019). More generally, flow rates such as saturated hydraulic conductivity are common parameters in environmental models,

and their scaling behavior should be considered (Zhu and Mohanty, 2002; Kumar et al., 2013b). Additionally, their anisotropic properties necessitate a dimension-dependent selection of upscaling operators (e.g., harmonic and arithmetic mean).

In this paper, we first present a working example to illustrate current challenges in the estimation of distributed parameter fields in environmental models. We then outline different ways to address this task and demonstrate how the new generic and agnostic `MPR` tool is designed and configured to meet the requirements. The section elucidating the configuration of MPR

is followed by a detailed description of how to interface MPR through a stand-alone executable, as well as through its API (application programming interface). We demonstrate its tight coupling to the hydrologic model mHM and its capabilities with regard to the reproduction of the original mHM model behavior. Furthermore, we demonstrate the versatility of the `MPR` tool by reproducing an open source dataset (EU-SoilHydroGrids (Tóth et al., 2017)) containing soil hydraulic properties derived from a set of TFs. The effects on long-term evapotranspiration from the coupling of MPR to state-of-the-art land surface models

(Noah-MP and HTESSEL) are also shown for an effective TF application and developing a regridding tool applicable to any model that requires distributed parameters (e.g., LSM or environmental models)

## 2 A minimal working example in environmental modelling

### 2.1 Objective

For demonstration purposes we define an objective that is commonly encountered in environmental modeling. For a hypothet-

ical inter-comparison project, the influence of different parameter estimation schemes for soil parameters is investigated for a given domain along with two different resolutions and three different model-specific grid types.

A common parameter present in many environmental models is soil porosity ($\theta_s$), which denotes the pore volume fraction of the total soil volume in the vadose zone. The SoilGrids (SG) dataset (Hengl et al., 2017) provides soil physical properties at a high resolution ($\frac{1}{480}^\circ$). From the extensive literature on pedotransfer functions (TFs for soil parameters) (Patil and Singh,

2016), we selected a TF for estimating $\theta_s$ based on bulk density, organic matter, clay, and sand content (Weynants et al., 2009). The southeastern United States, which includes the state of Florida, was chosen as the domain of interest due to high heterogeneity in the physical properties of the soil in this region. We selected different grid layouts that are often used in different modeling disciplines. These are regular rectangular grids generally used in distributed environmental modeling (Ma et al., 2017; Zink et al., 2017), icosahedral grids representing the group of geodesic grids increasingly used in the Earth System

Science community (Zängl et al., 2015; Skamarock et al., 2012), and polygons or hydrologic response units (HRUs) often used in hydrology or the soil sciences (Wellen et al., 2015).



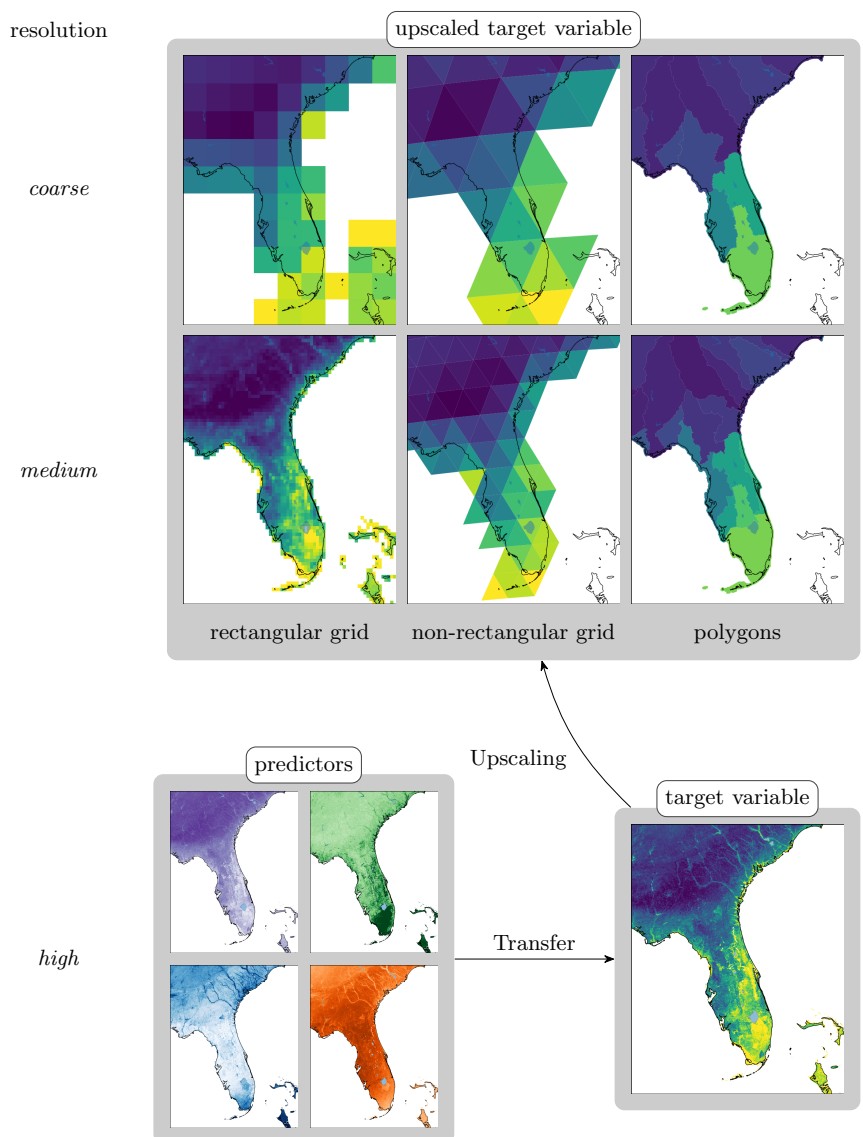

**Figure 1.** Example application of MPR deriving porosity based on SoilGrids250m (Hengl et al., 2017) variables (bulk density, organic matter, clay and sand content) in Florida, USA

The exemplary workflow depicted in Fig. 1 shows the high-resolution predictor variables (shown in the lower left corner) that are passed to the transfer function (TF) to derive the resulting field porosity (lower right corner) at the predictor resolution. It exhibits considerable heterogeneity in various gradients between the southeast and northwest regions of the domain, coastal and inland areas, riverbeds, and mountainous areas. The target variable is then upscaled to six different spatial grids at different




resolutions. Regular rectangular grids with resolutions of $1/8$ and $1°$ are shown in the left column of the top panel of Fig. 1. The lower grid is the same as the one used for the North American Land Data Assimilation System 2 forcing dataset (Mitchell et al., 2004). The latter dataset covers the domain of the conterminous United States (CONUS) and has been used for different LSMs (Xia et al., 2012). The two icosahedral grids specified by identifier R02B04 and R02B05 (Fig. 1 the center column

in the top panel (Zängl et al., 2015)) can be used for the configuration of LSM JSBACH (Mauritsen et al., 2019). Finally, polygons denoting the WBDHU4 and WBDHU6 domains (Fig. 1 right column in the top panel) and the realizations of the National Watershed Boundaries Dataset (of Agriculture-Natural Resources Conservation Service , USDA-NRCS) can be used for HRU-based (Flügel, 1995) models such as SWAT (P. W. Gassman et al., 2007) or PRMS (Leavesley et al., 1983). The figure demonstrates how different grids represent compromises between conservation of subgrid heterogeneity and reduction

of spatial complexity. Although the parameter gradients in the valleys in South Carolina in the parameter fields of the $1/8°$ rectangular grid are still visible, they are not visible in the other grids. The reduced number of grid cells allows for a lower computational load during model run times (simulations).

This parameter estimation routine, which is followed by simulations of the distributed model, might then be used iteratively by a calibration routine in order to optimize TF parameters. These are global parameters of the TF whose application to the

predictors is used to derive the effective distributed model parameters. Next, a workflow to derive these parameters in an efficient and consistent manner is presented.

### 2.2   Options for parameter estimation workflow

There exists a range of different options for the workflow described in the previous section, which we briefly describe here. A few existing software tools and GCM couplers can be used to perform the two key steps of applying a TF and remapping a

multi-dimensional grid onto an another:

1. In climate sciences, the command line data processing tools `cdo` (Schulzweida, 2019), `nco` (Zender, 2008) or `TempestRemap` (Ullrich and Taylor, 2015) are commonly used. `cdo` can assign functions to be evaluated and is applied on existing variables (`expr` operator). Its remapping commands are based on the original SCRIP implementations (Jones, 2010) that were translated in the $C++$ language and supplemented by the remapping schemes of the nearest neighbor and largest-

area fraction. `nco` is very similar to `cdo` and can perform both application of TFs (`ncap2` operator) and remapping (`ncremap` operator). `TempestRemap`, however, it serves solely as a regridding library.

To perform parameter estimations using one of these tools, we need to establish a stack of calls to the `expr` operator and `remap`. This can be achieved either directly or from a scripting language, for which wrapper libraries are provided. To improve the usability of the approach, the setup would need a wrapper library on its own to manage file-I/O, parameters,

coordinates, transfer functions, and upscale operations. One advantage of the developed MPR tool is that the entire workflow is described in a single configuration file.

2. Numerous couplers of Earth system models (ESMs) contain routines for the remapping of variables (e.g., ESMF (Collins et al., 2005) with `RegridWeightGen`, Atlas (Deconinck et al., 2017) or OASIS (Craig et al., 2017) among others). For





example, parts of the ESMF library can be compiled to the ESMF `RegridWeightGen` application, which readily interfaces with multiple netCDF-based grid formats and performs a number of different remapping algorithms. In addition, LFRic (Adams et al., 2019) and Atlas also provide a data model explicitly designed to support HPC applications. All of the aforementioned software tools and couplers expose their API and are publicly available and actively maintained by their respective communities.

Coupling libraries perform a multitude of tasks and thus have a codebase of considerable size. The installation procedure requires multiple dependencies on third-party software and is extremely demanding. Apart from `RegridWeightGen`, access to the correct sections of the API for remapping and function evaluation is not easily attainable. Accurate imitation of MPR functionality would require a setup similar to that of `cdo`, as well as establishment of a wrapper that effectively executes a set of commands or directly accesses the backend routines.

3. The processing of polygon-based data from shapefiles or geodatabases is often conducted using Geographic Information Systems (GIS) such as `QGIS` (team, 2020) or `ArcGIS` (ESRI, 2020). The support of netCDF-based data has been introduced in recent versions of `QGIS` by the `MDAL` library (Ltd., 2020). GIS possesses a large inventory for handling spatial data and associated visualizations.

After launching the target program, for example, `QGIS`, the predictor dataset needs to be loaded and the TF must be applied on the variable through the field calculator. A new layer is created through application of the appropriate spatial interpolation plugin. The exporting of workflow steps to Python is available for automatization.

## 2.3 Motivation for a new software tool

To the best of our knowledge, none of the aforementioned GCM couplers and GIS support out-of-the-box TF and upscaling applications. Although `cdo` and `nco` support dynamic evaluation of algebraic expressions, their implementation is cumbersome, bound to heavy disk I/O, and has a long runtime because of the dynamic evaluation of the transfer function. Additionally, not every tool supports remapping between polygons (e.g., basin boundaries or hydrological response units). `cdo` poses further restrictions on variable dimensions (conventionally, it supports only specific X, Y, Z and T axes). The support of dimension-dependent upscaling operators is not available for remapping tools (e.g., vertical dimensions must be handled differently than horizontal dimensions in aggregation of soil horizon specific parameters).

Previous software implementations of the MPR framework (Samaniego et al., 2010; Mizukami et al., 2017) have limited applicability as a result of hard-coded parameters, TFs, and upscaling operators.

The aforementioned restrictions inspired the motivation to rewrite `MPR` from scratch.

## 2.4 Nomenclature, conventions and general design

The API is closely built around the netCDF file format (version > 4.4.1) (Unidata UCAR, 2020). The resemblance of the netCDF format is motivated by its widespread use in environmental modeling and its paradigm of being scalable, portable, and self-describing. A file typically contains numerous multidimensional variables. Each dimension of each variable is an array of





monotonically ordered integers and can be shared among multiple variables. Each variable and dimension has a name, and the referencing of dimensions is done using these names. The CF (Climate and Forecast) convention (Eaton et al., 2017) further defines a coordinate variable as a one-dimensional variable sharing the same name as its associated dimension. Its values are ordered monotonically and missing values are prohibited. Auxiliary coordinate variables also contain coordinate information, but their names differ from their dimensions. Attributes containing meta information can be added to various objects such as variables, dimensions, and the file itself.

MPR adopted this concept and nomenclature for its data structures while imposing a number of additional restrictions. MPR variables are called `Data_Arrays`, as variables is a very generic term. We drop the concept of dimensions in favor of coordinate variables. This is necessitated by the fact that each cell needs to be explicitly bounded along its dimensions to avoid ambiguities during upscaling. Coordinate variables are referred to as `Coordinates`. `Data_Arrays` need to point to instances of `Coordinates` defining their extent. We also assume that each one-dimensional coordinate variable, in principle, represents not a point but an interval (or cell) and that adjacent intervals are contiguous for 1-dimensional coordinates. MPR supports the boundary variables as defined by Eaton et al. (2017), and is generally able to handle two-dimensional auxiliary coordinate variables as well. Thus, MPR accepts either one-dimensional or two-dimensional auxiliary coordinate variables. Users can set custom string attributes to coordinate variables for creating a self-describing output file.

## 3 Salient configurations of the developed MPR tool

### 3.1 Interfacing a standalone executable MPR

We show an example configuration in Fig. 2 to derive a target variable that is similar to what is shown in Fig. 1 (lower center map in top panel).

The configuration is performed in a Fortran-native, hierarchically organized namelist format (comparable to .json, .yaml, .ini, etc.). MPR has three required sections and two optional sections. Users can enter the minimal required information in a flexible and intuitive manner. The required sections `Main`, `Data_Arrays`, `Coordinates`, and the optional sections `Parameters` and `Upscalers` are designed as arrays of respective objects whose specific properties are set using the correct list index.

The term `Data_Arrays` refers to any n-dimensional variable and serves as a generic term for predictors and target variables. In lines 1ff. in Fig. 2, there are four `Data_Arrays` specified: the first three are read directly from the file, while the fourth is calculated from the former. The property `from_data_arrays` specifies the array of predictors to be used. They also appear in the TF equation that is supplied as a string in `transfer_func`. It follows the Fortran syntax for operators, brackets, and some elemental mathematical functions (see Appendix Table E1 for a list of possible operators). Users can use any parameter defined in `Parameters` in a TF. The TF string is not dynamically evaluated during execution because it leads to unnecessarily high computational run times. Additionally, the effort required to modify the Fortran source code each time a new function is used would substantially diminish the user-friendliness of the MPR tool. Instead, a Python preprocessor script is implemented that interprets and adds the transfer function from the namelist and modifies the source code accordingly. At





```
1   &Data_Arrays
      ! predictor variables
      name(1) = 'sand'
      from_file(1) = './input/sand_content.nc',
      name(2) = 'om'
6     from_file(2) = './input/organic_matter.nc',
      name(3) = 'bd'
      from_file(3) = './input/bulk_density.nc'
      ! target variable
      name(4) = 'k_zero'
11    from_data_arrays(1:3,4) = 'sand', 'bd', 'om'
      transfer_func(4) = 'exp(a5 + c5 * sand + d5 * bd + e5 * om) /
          unit_conversion'
      target_coord_names(1:2,4) = 'soil_layers', 'target_grid'
      upscale_ops(1:2,4) = '-1.0', '1.0'
      to_file(4) = .true.
16  /
    &Parameters
      ! global parameters
      parameter_names(1:4) = 'a5', 'c5', 'd5','e5', 'unit_conversion'
      parameter_values(1:4) = 1.9582, 0.0308, -0.6142, -0.1566,
          8640000.0
21  /
    &Coordinates
      ! specifications for the vertical target coordinate
      coord_name(1) = 'soil_layers'
      coord_from_values(1:4,1) = 0.1, 0.4, 1.0, 2.0
26    coord_cell_reference(1) = 'end'
      coord_from_values_bound(1) = 0.0
      ! specifications for the horizontal target coordinate
      coord_name(2) = 'target_grid'
      coord_from_file(2) = './input/target_grid.nc'
31    coord_sub_dims(1:2,2) = 'x', 'y'
    /
    &Main
      coordinate_group(1:3,1) = 'x', 'lon', 'target_grid'
      coordinate_group(1:3,2) = 'y', 'lat', 'target_grid'
36    coordinate_group(1:3,3) = 'z', 'depth', 'soil_layers'
      out_filename = '/output/OutputFile.nc'
    /
```

**Figure 2.** Example *mpr.nml* file for calculating $k_0$ for the R02B05 ICON grid following Weynants et al. (2009) using the SoilGrids (Hengl et al., 2017) dataset.

runtime, simple search and replacement routines translate the string into a unique function ID that is checked against all unique function IDs in the source code.

The target coordinates and associated upscaling operators are set with `target_coord_names` and `upscale_ops`. Upscaling operators are real numbers provided as strings. They specify the p parameter of the Hölder Mean/Geometric Mean (Sykora, 2009) (when entered as a real number). Alternatively, the subgrid minimum, maximum, sum, variance, or the value with the largest area fraction can be used when entered as a string. A full list of all possible operators is provided in Appendix Table E2. A flag `to_file` can be set to signal the Data_Array to be stored on the disk.

All parameters referred to from any TF can be specified by name and value in the `Parameters` section (l. 18ff. in Fig. 2). The TF for the fourth Data_Array requires multiple parameters (e.g. `a5`). Accordingly, more parameters can be set and reused in multiple TFs, while users should avoid naming duplications with TF operators or Data_Arrays. Parameters usually encompass constants and variable parameters subject to optimization, and as such, the `Parameters` section can also be read from a





seperate file containing only the parameters subject to optimization. This file is optional and its parameters replace previously
read parameters in the case of duplicates.

The target coordinates are specified in the `Coordinates` section (l. 26ff. in Fig. 2). The user explicitly specifies the boundaries of the soil layers by values in this example (`coord_from_values`). These refer to the stagger of each cell or horizon (`coord_stagger`). In this case, the first cell does not have a neighboring cell boundary, and as such the user must specify the coordinate boundary to provide a start value (`coord_from_values_bound`). A two-dimensional coordinate serving as the
target grid is read from the file (`coord_from_file`). Its associated dimensions are specified through `coord_sub_dims`. Alternatively, coordinate values can be specified from a range if the user provides values for start, step, and count.

The `Main` section (l. 37 ff. in Fig. 2) configures general information. The link from the target coordinates to the respective source coordinates during upscaling is constructed through the `coordinate_groups` entries. During upscaling, each source coordinate needs to receive a target coordinate. If the source coordinate shares the same `coordinate_group` as one of the
target coordinates, upscaling is performed for that coordinate pair. Users can enter an arbitrary number of groups, although a dimension system with X, Y, Z, and T is most commonly defined. Finally, the path to the output file is set.

A more exhaustive description of the aforementioned configuration can be found in online documentation.

## 3.2 Interdependence of parameters

Another example configuration highlighting the capability of MPR for creation of netCDF variable graphs is shown in Fig. 3,where
each node is a Data_Array and the links consisting of TFs and/or upscaling operations. It visualizes the dependency graph for two of the parameters required for the mHM in its standard configuration. The blue ellipses denote the predictor variables used. While `land_cover` is a 3-dimensional array with coordinates, year, latitude, longitude (t1, y, x), `lai_class` is a 3-dimensional array with coordinates, month of year, latitude, and longitude (t2, y, x). The TF for the model parameter `PET_LAI_corr_factor` requires both LAI and land cover information. Both predictors need to be broadcast to a
4-dimensional array (t1, t2, y, x), and so temporary arrays are created. The TF for the model parameter `Aerodyn_resist` requires information on canopy and wind measurement height, while the wind measurement height is derived from the former alone. For the calculation of `canopy_height`, a max-normalized LAI is needed for each cell. The Data_Array contains this information, and then its broadcast variant `LAI_max_t2_4D` will be the same shape as the other predictors of the TF responsible for producing `canopy_height`. The two model parameters highlighted by the red ellipsis are finally upscaled to the
target model resolution with coordinates modeling year, month of year, low resolution latitude, and low resolution longitude.

## 3.3 Interfacing MPR library

While the previous section introduced the use of MPR as a standalone wrapper, we anticipate a tight coupling of the MPR code to the main modeling code. We intend to impose maximum reusability of the MPR API and ease its implementation into other libraries. Top-level objects such as Data_Arrays or Coordinates can be reused multiple times and can also be written to and read
from the disk as requested. The Fortran API is used here for this purpose, which is based on the object-oriented programming paradigm and exposes four main objects (derived type in Fortran), which are also present in the namelist configuration (see

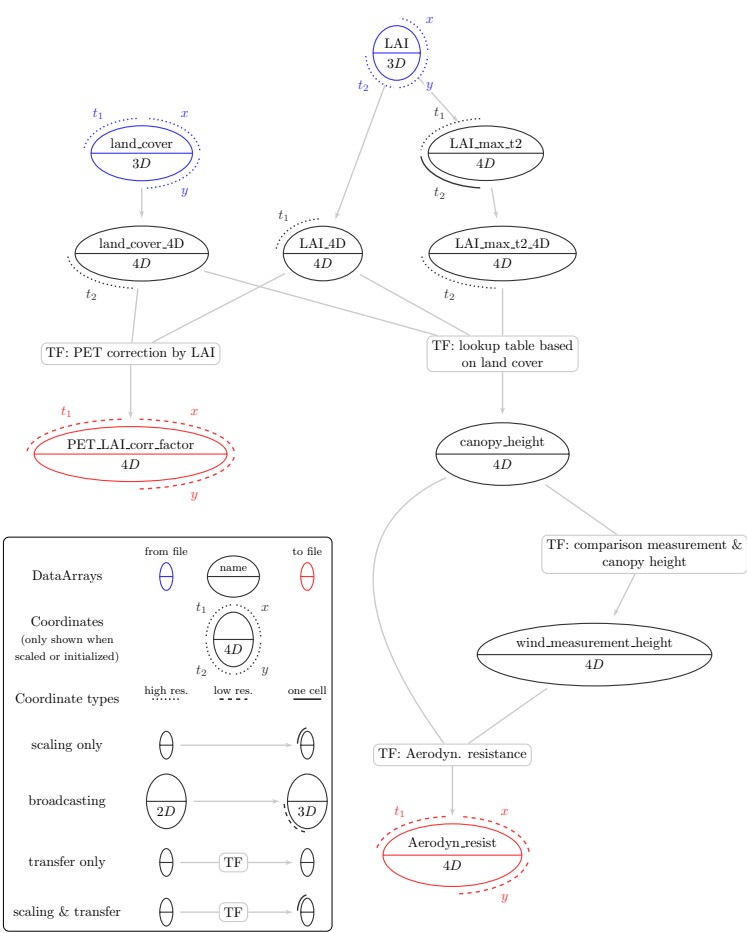

**Figure 3.** Dependency graph of two of the mHM parameters in their standard configuration.

previous section): `Coordinate`, `Parameter`, `Data_Array` and `CoordUpscaler`. After initialization, these types are stored in global arrays which allow for cross-referencing with other types. They are briefly described in this section and additional detail can be found in the online documentation.

### 3.3.1 Coordinate type

One central derived type in MPR is the `Coordinate`. First and foremost, it stores the boundaries of its $n$ cells. In the case of a (one-dimensional) coordinate variable, each cell has two boundaries ($v = 2$) and as cells are contiguous, the boundaries do not need to be stored in a $(v, n)$-shaped array but in an $(n + 1)$ array. This property is termed as `boundaries1d`. In the case





of a two-dimensional auxiliary coordinate variable, the number of boundaries/edges can vary. To obtain a general formulation,

we stored the boundaries of both dimensions in an $(v, n)$-shaped array. These properties are termed `boundaries2dDim1` and `boundaries2dDim2`. Additionally, the cell centers are stored in `centers2dDim1` and `centers2dDim2`.

### 3.3.2  Parameter type

`Parameters` are objects with names and numerical values assigned to them.

### 3.3.3  Data_Array type

The main derived type is the `Data_Array`, which can be read directly from an existing netCDF file or computed from other `Data_Arrays` and/or `Parameters` using TF and upscaling operators. It stores multidimensional data which must be passed to multiple other routines. In order to have a common sparse data container, non-masked cells are stored in a one-dimensional array `data` with type real, regardless of the number of underlying dimensions. A Boolean mask of the data is stored in a flattened, one-dimensional array, `reshapedMask`. A one-dimensional array pointing to its associated `Coordinates` is set

as `coords`. This holds the shape information of the original uncompressed data. These core properties, `reshapedMask`, `data`, and `coords`, are pointed at from within the wrapper type `InputFieldContainer`. It is used for referencing the core properties of the `Data_Array` and are usually passed as an argument for the `TransferHelper` type, which is described in the next section.

### 3.3.4  Application of Transfer Functions

The type `TransferHelper` is intended to be used for the initialization of `Data_Array`, which originates from either a file or from other `Data_Arrays` via TFs. During its initialization, it performs checks on the passed `InputFieldContainer`, checks their `Coordinates`, optionally concatenates `Data_Arrays`, and optionally adapts their masks. After these checks, the TF is applied. TFs are designed as a subroutine accepting an arbitrarily-sized list of `InputFieldContainer` (its `data` property is accessed only) and `Parameters`, respectively. An abstract interface for TFs thus allows for a variable number of

predictor arrays and parameters. A one-dimensional array is returned. We provide a template for a subroutine containing a TF so that users can modify the source code and implement their own TFs (see Appendix Fig. D1). This enables the integration of more complex mathematical formulations such as artificial neural networks or support vector machines. Based on the majority of TFs used in existing LSMs (Van Looy et al., 2017), we anticipate users primarily using TFs that employ common mathematical operators such as multiplication and division. Such equations can be automatically parsed from configuration

files and inserted into the Fortran source code using a Python preprocessor, which is described next.

Each TF string occurring in a namelist is translated into a unique identifier. For example, string `exp(a5 + c5 * sand + d5 * bd + e5 * om) / unit_conversion` (Fig. 2) can be set in the configuration file and will then be analyzed and processed by a parser routine. MPR replaces all parameters (`a5`, `c5`, `d5`, `e5`, `unit_conversion`), and variable names (`sand`, `bd`, `om`) by identifiers, translating the resulting string (`exp (p1 + p2 * x1 + p3 * x2 + p4 * x3) /`





p5) into a unique function identifier (`exp_bs_p1_pl_p2_ti_x1_pl_p3_ti_x2_pl_p4_ti_x3_be_di_p5`). This identifier represents the exact mathematical function that can be used for multiple applications with different `Data_Arrays` and `Parameters`. The number of TFs contained in the source code is thereby reduced, and duplications of TFs are avoided. TFs support multiple operators such as scalar numeric expressions (e.g., $*$, $/$, $+$, log,...), trigonometric functions, relational operators ($>=$, $==$, ...), logical expressions, and constructs using if and where expressions (see Appendix Table E1). The

resulting identifier is checked against existing identifiers in the source code.

### 3.3.5 API for Upscaling

For the upscaling step, two `Coordinates` sharing the same dimensions are compared. For each cell of the target grid, the underlying $n$ source cells (`subcells`) are stored in an `ids` array and their relative contributing area is stored in a `weights` array. These properties are contained in the type `CoordUpscaler`. They can be initialized from existing remapping weights

stored in the netCDF file format following the SCRIP convention (Jones, 2010). By default, first-order conservative remapping is used for weight calculation, so that the integral of the values is preserved in the presence of missing values. Two-dimensional auxiliary coordinate variables are mapped using a simple algorithm that only checks that the cell center of the source cell is within the target cell boundaries and assigns equal weights to each source cell. In the future, more sophisticated remapping schemes for two-dimensional variables will be implemented.

The upscaling of a `Data_Array` is conducted through the wrapper type `UpscaleHelper`, which is also a property of the `Data_Array` type. It consists of a one-dimensional array pointing to the source coordinates `sourceCoords` and target coordinates `targetCoords`, as well as the upscale operator names for each target coordinate `upscaleOperatorNames`. The `Upscaler` type performs multiple checks on the source and target coordinates of `Data_Array`. If applicable, it transposes or broadcasts the `Data_Array` if the order of source and target coordinates does not match. Multiple `CoordUpscaler`

instances can be combined with an aggregated `CoordUpscaler` object, effectively combining the weights and subcell IDs when the user specifies the same upscaling operator for multiple coordinates of the target grid. The upscaling operation is then executed separately for each group of target dimensions using the same upscaling operator. There are a number of standard upscaling operators, such as the minimum, maximum, or weighted generalized mean function, which employ the power parameter (see Appendix Table E2). Users can easily add another upscaling function to the source code as long as it effectively

aggregates the variability of the subcells into one value (see Appendix Fig. D2).

### 3.4 New features of current MPR release

The current set of features encompasses the functionality of the previous version of MPR as used in the mHM source code (Samaniego et al., 2019). This implementation lacks some key features that are highlighted by a comparison of Fig. 1 with Fig. 1 in Kumar et al. (2013a). First, the new MPR library is modularized and refactored, so it does not depend on mHM

and its integrated optimization algorithms. Second, the original mHM implementation required grids to be rectangular, and the target resolution was a multiple of the source resolution. Finally, predictors, TFs, upscaling operators, and target grids can now be freely chosen and recombined in an MPR configuration file.





Furthermore, it allows for a modular and flexible configuration of parameter estimation without common preprocessing steps of the input files (resorting coordinates, renaming variables, adding missing meta information, applying unit conversion, etc.).

Use of MPR is easy and intuitive, especially in the formulation of TFs, defining new coordinate variables, and assigning them to variables. The initialization of coordinate variables can be performed dynamically depending on existing coordinate variables (e.g., by using its bounds). The user can freely and flexibly enter simple regression-based TFs in a semi-automatic manner. More sophisticated functions, such as artificial neural networks or support vector machines, can be easily coupled to the code. The same holds true for upscaling operators. We allowed MPR to be easily integrated in workflows (e.g., auto-calibration) or

have an API called from an external code.

Coordinate variables can be split and combined, which enables users to set coordinate-dependent parameter values (e.g., for certain horizons along a soil profile). The order of coordinate variables for individual variables can be changed. MPR supports up to 5-dimensional variables without restrictions on their kind. Different upscaling operators can be chosen for each coordinate to be upscaled. Intermediate variables can be reused for the creation of other variables, allowing the creation of

complex graphs of parameter interdependencies. Parameters can also be reused for multiple TFs.

## 4 Applications of MPR

### 4.1 Verification of MPR against previous version in mHM

The core objective of the new MPR tool was to reproduce the same model behavior as the original implementation of MPR in the mesoscale hydrological model (mHM) (Samaniego et al., 2019), which we refer to as mHM-MPR here. The model

description, its code modifications, and the MPR configuration can be found in the Appendices B1 and C1.

The comparison of the new MPR tool coupled to mHM with mHM-MPR shows that they yield the same simulation results within a tolerance of 0.1%, which can be attributed to floating point precision deviations after the conversion of the input file format from text files to netCDF. In a default model configuration of the Moselle river basin in Central Europe, the coupled version reproduced the same model parameter arrays. Consequently, this leads to differences in the hydrograph of the basin

outlet within $10^{-5} m^3\ s^{-1}$, as compared to the previous implementation.

### 4.2 Reproducing EU-SoilHydroGrids dataset with MPR

We selected the EU-SoilHydroGrids (SHG) dataset (Tóth et al., 2017) as it is relevant to the Earth System Modeling community and is publicly available. The same holds true for its predictors and the TFs used. We reproduced the aforementioned dataset and showed the salient seamless spatial scaling feature of the MPR tool. The dataset description and its configuration can be

found in the Appendices B2, C2 and the MPR configuration in the Supplements.

Fig. 4 shows the spatial distribution of $K_s$ (soil saturated hydraulic conductivity) on a logarithmic scale for different resolutions and data sources at a depth of 15cm with values ranging across three orders of magnitude from $10^{-6}$ to $10^{-3}$ m s$^{-1}$. Each row of the plot shows grids with the same spatial resolution, and each column shows the data sources and processing schemes.



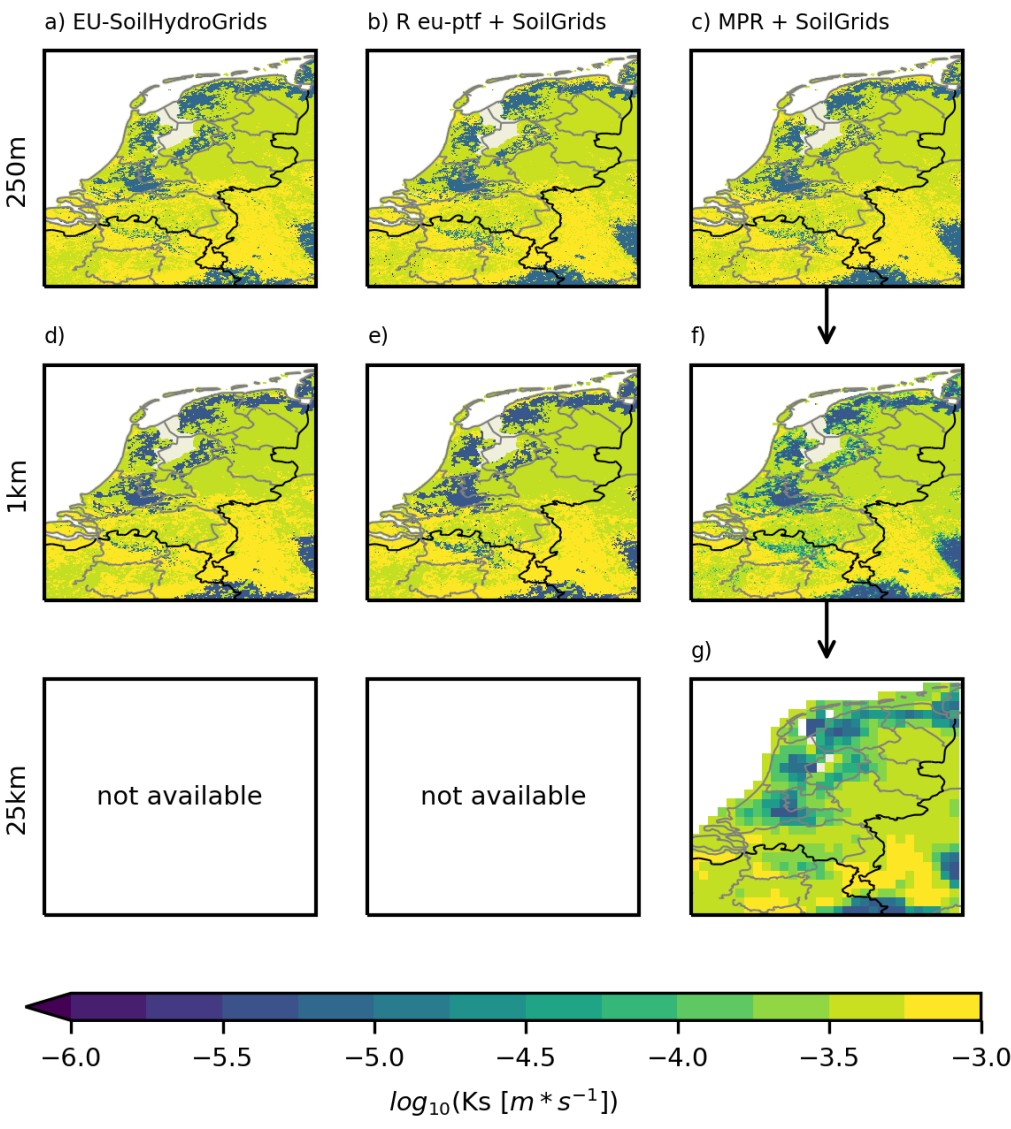

**Figure 4.** Saturated hydraulic conductivity values for the Netherlands generated for different scales and different tools: a) EU-SoilHydroGrids (Tóth et al., 2017) at 250m b) TF16 (Tóth et al., 2015) applied on SoilGrids (Hengl et al., 2017) using R package eu-TF at 250m c) TF16 (Tóth et al., 2015) applied on SoilGrids using MPR (Hengl et al., 2017) (Weynants and Tóth, 2014) at 250m d) EU-SoilHydroGrids (Tóth et al., 2017) at a 1km e) like b) at 1km f) TF16 (Tóth et al., 2015) applied on SoilGrids250m and scaled to 1km using MPR g) TF16 (Tóth et al., 2015) applied on SoilGrids250m and scaled to 25km using MPR.

The values were estimated using a decision tree with 21 leaves (Tóth et al., 2015, Table S1, model 16). When we attempted
to reproduce the SHG values (Fig. 4a) with the recommended procedure in Tóth et al. (2015) using an R-library (Fig. 4b), we





obtained a bias of up to 10% in comparison to a). An investigation of the project webpage revealed that the SoilGrids (SG) dataset is subject to frequent updates depending on the arrival of new source soil profiles. We could not verify the exact version of SG used to create the SHG, but we assume that it was different than the one we used. The high bias observed is a result of the decision tree, as small deviations in predictor values might result in use of a different branch. An additional source for the
observed bias could be the projected coordinate system used for the SHG, in contrast to SG being available in the geographical WGS84 coordinate system.

When using MPR to apply the TFs to the SG dataset (Fig. 4c), we obtained a mean relative bias below 0.06 % compared to b), which was only due to rounding errors of the global parameters. SHG and SG are also available at a 1 km resolution, where variables are aggregated using the Geospatial Data Abstraction Library (contributors, 2019) `average` method (Fig. 4 d) and
e)).

In accordance with the MPR framework, we derived the 1 km $K_s$ field by averaging the high-resolution SG data at 250 m (Fig. 4c). Because the selected TF is based on a decision tree algorithm, there exist only some discrete values in Fig. 4d) and e), whereas spatial averaging increases the variability of the derived parameter field. The capability of MPR in flexible aggregation of the data is shown in Fig. 4 (g)). A resolution of 25 km was arbitrarily selected as the target resolution, as specific resolutions
(e.g., 1 km) are typically not needed by users. By a mere change of one number in the configuration file, it is possible to scale the variable at interest to every user-defined resolution. This analysis highlights the capability of MPR in reproduction of environmental parameters and generation of outputs that meet specific user requirements.

### 4.3 Application with land surface model Noah-MP

We used the land surface model (LSM) Noah-MP (Niu et al., 2011) as an example to showcase the capability of MPR in
coupling with state-of-the-art distributed environmental models. The model description and configuration can be found in the Appendices B3, C3, the MPR configuration in the Supplement. We found that the inherent parameter uncertainty that occurs when choosing a TF and an upscaling operator for the soil parameters of the model eventually leads to differences in long-term mean annual evapotranspiration (ET) flux, up to 20% (Fig. 5) in relation to its default setup.

Fig. 5 shows the absolute values of the default model parameters $K_s[m * s^{-1}]$ (log10-transformed) and $\theta_s[-]$ and the re-
sulting long-term annual ET flux $ET[mm\,year^{-1}]$ in the first row (maps 1a-1c). The following rows (2-4) show the relative differences (in %) of the field with respect to the default simulation in row 1. The relative differences observed when using the MPR approach with the same soil dataset, same TF, and subsequent spatial upscaling are shown in the second row (subplots 2a-c). Larger differences occur in regions with considerable subgrid heterogeneity in soil texture, where the hydraulic parameters associated with the mean textural information of the dominant class do not represent underlying variability. Absolute
differences of more than 5% for both parameters occur in Florida, parts of Nebraska, and parts of the Southwest. The average difference is -1.6 and -1.3 % for both parameter fields, and a greater variability can be found for $\theta_s$. The difference for the long-term ET flux is -3 % on average, with pronounced low values of less than -10 % in the aforementioned regions. The lower ET fluxes are due to the accumulated effects of lower porosity and lower hydraulic conductivity, which leads to decreased storage capacity and capability to meet evaporation water demands. It is important to keep in mind that these changes stem solely

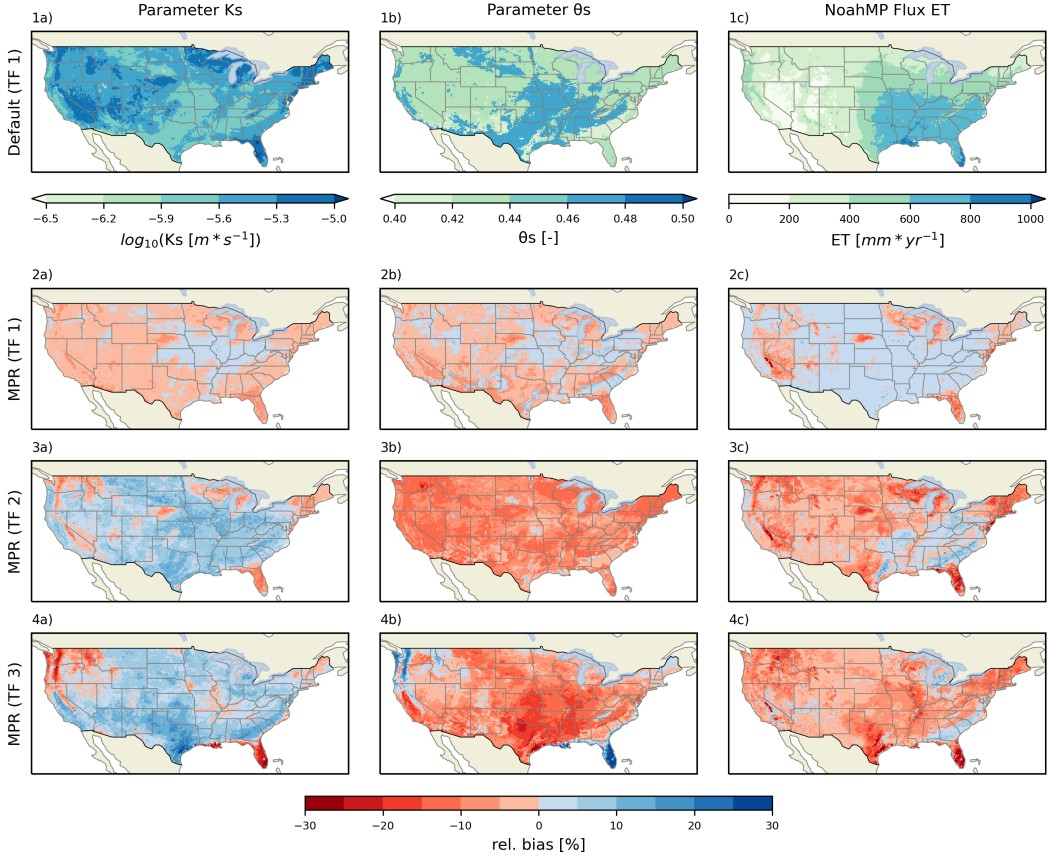

**Figure 5.** Grid plot showing maps of parameters and simulation results for Noah-MP. Columns denote: a) Soil parameter $K_s$ (saturated hydraulic conductivity) at the soil horizon between 0.1 and 0.3 m b) Soil parameter $\theta_s$ (maximum soil moisture content) at the soil horizon between 0.1 and 0.3 c) Mean annual evapotranspiration values of Noah-MP. Rows denote: 1) Standard Noah-MP setup using the dominant USDA soil class based on SoilGrids (Hengl et al., 2017) and lookup table based on the TF from (Cosby et al., 1984) 2–4) relative differences $((MPR - default)/default)$ in % of parameters and simulation results using an MPR setup with a TF from Cosby et al. (1984), Saxton and Rawls (2006), and Vereecken et al. (1989, 1990), respectively.

from using non-classified continuous soil textural information and aggregation of subgrid parameters, in contrast to using the dominant soil type. In other words, variation originates from different methods of handling sub-grid variability.

In the next step, we replaced the TF in MPR with another continuous TF based on the linear regression of the predictors of sand content, clay content, and organic matter (Saxton and Rawls, 2006) (maps 3a–3c). This TF is also available within Noah-MP version 4.0. This leads to $K_s$ values that are on average 4.7 % higher than in the default setup. Application of the TF

results in a decreased $\theta_s$ over the majority of the domain, with average values around -8.6 %. The ET fluxes exhibit a decrease for most of the CONUS in comparison to the default setup (on average -4.0 % with maxima in the Appalachians of 18.4 % and minima in Florida of -27.3%) .





Yet another spatial pattern of parameter values is produced by application of the TF of Vereecken et al. (1989) for $\theta_s$ and Vereecken et al. (1990) for $K_s$. TFs use the predictors of sand content, clay content, bulk density and organic matter.

While the overall relative differences in comparison to the default parameter distribution of $K_s$ are positive, there are negative differences of approximately -25 % in the northwest areas of the CONUS domain, in Florida, and along the east coast between Louisiana and Virginia. An inverse signal can be observed for the parameter $\theta_s$. The overall decrease in ET that predominates in Texas, Florida, and the northern Rocky Mountains highlights the non-linear dependence of ET on the modified soil parameters as a new spatial pattern occurs. However, the results obtained must be assessed critically. Significant differences in parameter

values in some areas indicate the limited applicability of the chosen TF. Indeed, high values of bulk density exist in parts of southern CONUS, and high organic matter contents are present in Oregon, Washington, and along the western coastline. The reported range of values (0.01 to 6.6 % for organic matter content and 1.04 to 1.23 $g/cm^{-3}$ for bulk density), upon which the regression for the TF was computed (Vereecken et al., 1989) is based on a small dataset of soil samples from Belgium. This does not cover the range of the values present in the chosen soil dataset (i.e., SG). The applicability of a given TF needs to be

evaluated in the context of the utilized soil database (Wösten et al., 2001).

The model bias for a standard Noah-MP configuration in comparison to the reference product FLUXNET has been assessed in a previous study (Ma et al., 2017). They observed a spatial pattern in the long-term bias of ET in regards to prescribed leaf area index (LAI), which showed similarities to the pattern seen in Fig. 5 in maps 2c, 3c, 4c. Although the setup of Noah-MP was not identical, MPR might serve as a valuable tool in addressing the problem of model bias through improved parameter

estimation.

Cuntz et al. (2016) investigated parameter sensitivity in simulated fluxes by Noah-MP. They reported a strong sensitivity of ET on the soil parameters $\theta_s$ and $K_s$, which is in accordance to our findings shown in Fig. 5. One limitation in their results is that they directly investigated grid cell-specific parameters. As a result, the dimensionality of the parameter space linearly scales with the area of the model domain. Thus, only a few catchments using a spatially constant scale factor could be used.

Using MPR, the number of TF parameters remains independent of the size of the model domain, which allowed us to conduct a more spatially comprehensive sensitivity analysis. At the same time, MPR requires TFs for every parameter, which indicates that the uncertainty due to the choice of TF cannot be neglected when conducting a sensitivity analysis.

### 4.4  Application with land-surface model HTESSEL

We used the land surface model HTESSEL (Balsamo et al., 2009) as an example to showcase the capability of MPR in cou-

pling with state-of-the-art distributed environmental models. HTESSEL is the land-surface scheme used within the integrated forecasting system developed at the European Center for Medium-Range Weather Forecasts (ECMWF). The model description and its configuration can be found in the Appendices B4, C4, the MPR configuration in the Supplement.

Similar to the application of Noah-MP (Fig. 5), we found differences in the long-term evapotranspiration (ET) flux of up to 15% over the Mississippi river basin (Fig. 6) when using different transfer functions to compute soil hydraulic properties.

Fig. 6 is organized in the same way as Fig. 5. In its default configuration based on SoilGrids (SG), there exist only five different soil classes over the entire domain. Through a lookup table, each class is assigned the values for the van Genuchten



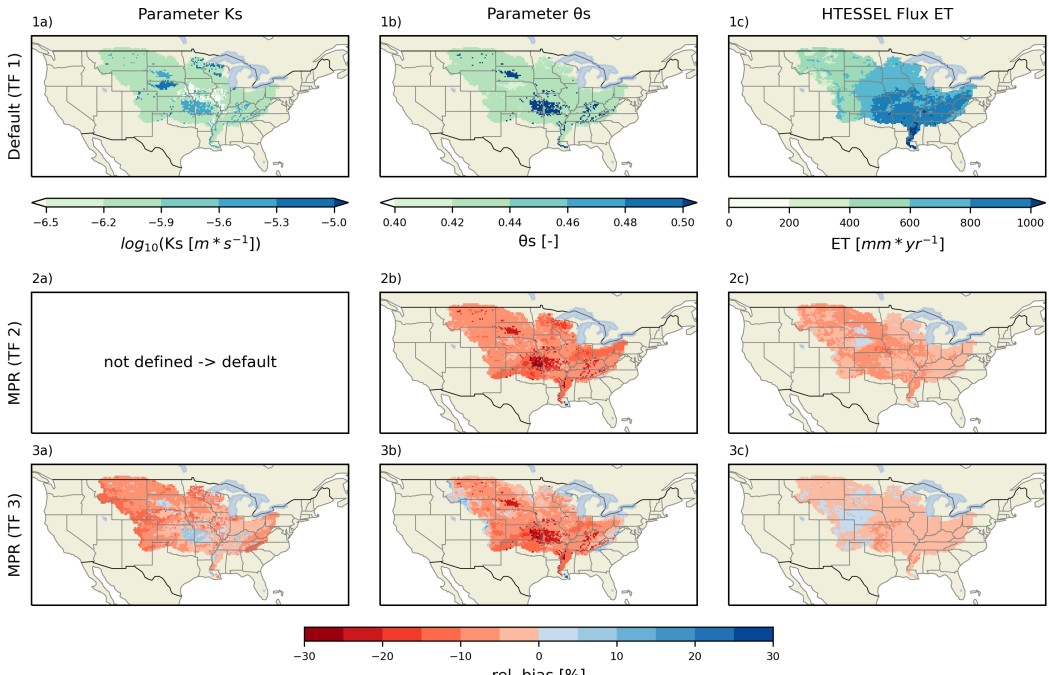

**Figure 6.** Maps of parameters and simulation results for HTESSEL. Columns denote: a) Soil parameter $K_s$ (saturated hydraulic conductivity) at the soil horizon between 0.07 and 0.28 m b) Soil parameter $\theta_s$ (maximum soil moisture content) at the soil horizon between 0.07 and 0.28 m c) Mean annual evapotranspiration (ET) values of HTessel. Rows denote: 1) Standard HTESSEL setup using the dominant soil classes based on the SoilGrids dataset (Hengl et al., 2017) and lookup table based on the TF from (Balsamo et al., 2009), 2-3) Relative differences $((MPR - \text{default})/\text{default})$ of parameters and simulation results using MPR based on SoilGrids (Hengl et al., 2017) with TFs from Zacharias and Wessolek (2007) (TF2) and Wösten et al. (2001) (TF3), respectively.

model (Genuchten, 1980) of the hydraulic conductivity curve (Equation 1) and moisture retention curve (Equation 2): $K_s$, $\alpha$, $n$, $l$, $\theta_r$, and $\theta_s$. It is worth mentioning that this lookup table was originally derived for the FAO 2003 soil map (, CBL).

$$\gamma = K_s \frac{\left[ (1 + (\alpha h)^n)^{1 - 1/n} - (\alpha h)^{n-1} \right]^2}{(1 + (\alpha h)^n)^{(1 - 1/n)(l+2)}} \tag{1}$$

$$\theta(h) = \theta_r + \frac{\theta_s - \theta_r}{(1 + (\alpha h)^n)^{1 - 1/n}} \tag{2}$$

The most common values are 1.16e-6 $ms^{-1}$ and 0.439 % for $K_s$ and $\theta_s$, respectively. These values correspond to the medium soil texture class in the lookup table. High $K_s$ values (up to 4.6e-5 $ms^{-1}$) can be found in Missouri, Kansas and the Nebraska Sandhills. Occurrences of high $\theta_s$ values of 0.52 (fine texture) can be found in Missouri and Kansas. Notably, both parameter





maps show the distribution of only two out of the six soil hydraulic parameters. This parameter selection is a dominant part of

all soil hydraulic properties that are relevant for simulated ET. Yet, it is not sufficient to show the highly non-linear relationship between soil hydraulic properties (i.e., hydraulic conductivity curves and moisture retention curve) and simulated ET at every location. Long-term ET values increases from 250 to 1450 mm per year along a gradient from the north-west to the south-east of the domain (Fig. 6 1c). We selected two sets of TFs from the literature to calculate soil hydraulic properties (Zacharias and Wessolek, 2007; Wösten et al., 2001). These TFs are applied to SG and can easily be implemented in MPR. After application of

TF 2 (Zacharias and Wessolek, 2007), the parameter values $\theta_s$ are reduced by about -9.0 % (Fig. 6 2b). The highest differences occur again in Missouri and Kansas (around -30 to -47 % for $\theta_s$). This TF does not contain Ks and the default Ks values are thus used. The change in $\theta_s$ reduces long-term ET flux by about -5.0 % (Fig. 6 2c). This can be expected because soil moisture storage (i.e., porosity) is generally reduced (Fig. 6 2b). In turn, the amount of water available for plant transpiration is limited. The usage TF 3 (Wösten et al., 2001) for the estimation of $K_s$ and $\theta_s$ reduces parameter values by approximately

-6 % for Ks and -7 % for $\theta_s$ on average over the entire domain. Applying TF 3 reduces porosity (Fig. 6 3b) in a similar order of magnitude compared to TF 2. Additionally, saturated hydraulic conductivity is reduced at most by 16% resulting in reduced ET in comparison to the default setup (Fig. 6 3c). This reduction is not as strong as that of TF 2 because the reduced $K_s$ values increase the water holding capacity of the soil.

By producing continuous fields of parameter values with MPR, increased/decreased $K_s$ values are found in regions with

low/high default values. Similarly, the highest reductions of $\theta_s$ are found in regions with the highest default values. This highlights that the MPR-derived fields reduce the amplitude of the parameter values, but substantially increase the spatial variability. MPR-derived fields make more use of the spatial information of the soil dataset and lead to more realistic spatial parameter fields. It is worth mentioning that the spatial patterns for changes in ET do not correspond to neither changes in parameter fields nor the spatial pattern of the default ET values. This indicates the complex interplay between ET and soil

hydraulic properties and calls for a deeper analysis of all MPR-derived soil parameters (i.e. also $\alpha$, $n$, $l$ and $\theta_r$ etc.).

### 4.4.1 Differences between HTESSEL and Noah-MP over the Mississippi river basin

There are several differences between the simulations conducted with Noah-MP and HTESSEL that go beyond the fact that these are two are based on different mathematical models. First, the default simulations compute different amounts of long-term ET (compare 1c in figures 5 and 6). Both maps exhibit a similar spatial pattern, but the long-term ET flux for HTESSEL

is approximately 20 % higher than that of Noah-MP. This might be due to the use of different forcing datasets NLDAS2 (Xia and NCEP/EMC, 2009) and ERA5 (ECMWF, 2019) for Noah-MP and HTESSEL, respectively. Xu et al. (2019) and Saxe et al. (2020) suggest mean precipitation of ERA5 is higher than in NLDAS2 in the study domain.

Second, the default soil hydraulic parameters show a different spatial pattern (compare 1a and 1b in Figures 5 and 6). The default setup uses lookup-table with a limited number of soil classes based on the TFs from Cosby et al. (1984) for Noah-MP

and HYPRES (1997) for HTESSEL. The estimation of the effective soil class follows the dominant class approach, which leads to a limited spatial variability of soil hydraulic properties for both models. It is worth mentioning that both lookup-tables were derived for other soil maps than the one used in this study (for example (CBL) for HTESSEL). Here, we applied both default





lookup-tables on the same dataset (SG) to rule out differences coming from different soil maps. While a decreased spatial variability especially for HTESSEL with only five active soil texture classes is found, the Cosby TF leads to a more consistent

spatial pattern for Noah-MP. Additionally to the spatial pattern, the parameter values themselves are also different for both models. $\theta_s$ for Noah-MP varies around 0.46, and HTESSEL again shows slightly lower values of approximately 0.44. $K_s$ is on average around 2.5e-6 for Noah-MP and much lower with 1.16e-6 for HTESSEL. These striking differences are in agreement with Samaniego et al. (2017), where a more exhaustive model comparison was performed. This study again highlights the need for a common protocol to assess parameter uncertainty in distributed models.

Third, using other TFs than the default ones leads to reductions in long-term ET around less than 10% in magnitude for both models (compare right column in figures 5 and 6). A similar magnitude of influence by varied soil parameters on ET has been reported previously by Livneh et al. (2015) for mHM in the Mississippi river basin. There is no consistent pattern between models in regard to where these changes manifest themselves. An example for that are the Nebraska Sandhills. While ET is generally increased by TFs in HTESSEL in this region, the opposite is the case in Noah-MP. Direct interpretation of the

interplay of soil parameters in the soil water hydraulics is easier with Noah-MP due to the simpler mathematical model for soil hydraulic parameterization. Noah-MP uses the Campbell parameterization to relate hydraulic conductivity to soil saturation (Campbell, 1974). In contrast, HTESSEL uses Mualem-van Genuchten parameterization (Genuchten, 1980), which leads to complex changes of moisture retention curves and hydraulic conductivity curves that are highly non-linearly impacted by changes in model parameters (not all of them shown here). Notably, models react to changes in (a limited number of) model

parameters for the case of ET fluxes investigated here. Larger changes can be found for other fluxes of the water cycle and sub-annual time scale (not shown).

In spite of demonstrated differences in model forcing, configuration and the process parameterization, MPR-derived parameter fields significantly changed long-term model output. The harmonization and reproducibility of parameter estimation across models through MPR opens up an avenue to a deeper understanding of the relationship between predictors, parameters and

model parameterization.

### 4.5 Limitations and outlook

In spite of the versatility of MPR for its use cases, we acknowledge that limitations do exist, some of which are technical while others concern supported features. The limitations are:

– The computational cost of MPR in large-scale applications and optimization applications is low due to a lack of paral-

lelization. This will likely change in future releases by utilizing a hybrid MPI and Open-MP parallelization.

– MPR cannot yet handle transformation of different projected or geographic coordinate systems.

– Additionally, so far MPR only accepts the netCDF format as input or output. However, if the user community requests an extension to include GRIB-based, polygon-based, or HDF5-based input data, it may be implemented in future versions.



MPR is not compatible with older compiler versions, such as the GNU gfortran 4.8 compiler, which is still common in
many systems because MPR uses an object-oriented programming approach and various features of the Fortran 2003 and 2008
standards. MPR was tested with compiler GNU gfortran versions > 7.3 the NAG Fortran Compiler version 6.2, and Intel ifort
version > 18.

We will invest further efforts in developing MPR so that scalability on high-performance computers (HPCs) and paralelliza-
tion is improved. The need for support of advanced and massively parallel regridding and interpolation capabilities will likely
lead to the integration of one of the existing remapping libraries used in GCM couplers in an upcoming version of MPR. The
coupling of MPR to the LSM HTESSEL by extraction of hard-coded and hidden parameters and the development of TFs in an
ongoing project will likely serve as a template that can be adapted for other models.

## 5    Conclusions

Parameter regionalization enables the creation of seamless parameter fields for complex distributed models that can otherwise
only be inferred through calibration or by default values, often obtained at inappropriate scales (Samaniego et al., 2017). MPR is
a framework that regionalizes parameters through the application of transfer functions and aggregations to any spatiotemporal
coordinate system. In this study, we introduced a complete rewrite of the MPR framework (Samaniego et al., 2010; Kumar
et al., 2013b) to overcome the limitations of previous implementations and comparable software (for example, Mizukami
et al., 2017). MPR is able to introduce new flexibility to mHM and other models accepting distributed parameters through
the support of multiple grid structures and a more flexible configuration of the parameter estimation process. It is capable
of reproducing effective parameter fields (Tóth et al., 2017) by applying TFs (Tóth et al., 2015), while also being able to
remap/upscale the parameters onto every modeling unit (rectangular grids, HRUs, arbitrary shapes) required by the model. We
demonstrated that parameter estimation not only exerts a strong influence on effective model parameter fields but also results
in modified evapotranspiration simulated by land-surface models, even when MPR is applied to only two sensitive parameters.
The same holds true for models that use a tiling approach for handling subgrid heterogeneity.

The superiority of the MPR approach toward standard parameter estimation approaches was first demonstrated by Samaniego
et al. (2010); Kumar et al. (2013b); Samaniego et al. (2017) and is now available for use with many other models. This is
possible because MPR is designed to flexible, modular, and as easy to use. We provide an API that users can easily modify
and that can be successfully tightly coupled to the hydrologic model mHM. As such, we invite implementation of further
TFs and upscaling operators in the other distributed modeling code. MPR provides a way forward in addressing many current
challenges regarding the estimation of distributed parameter fields in the Earth System Model community (as postulated by
Van Looy et al., 2017), such as coupled parameterizations and TF validation in large-scale applications, among others. It
serves as a protocol for systematic development of new TFs and aggregation schemes or upscaling approaches. As such, it
makes the whole process of parameter estimation transparent and reproducible. It can easily produce time or process-dependent
parameters (e.g., tillage systems, swell/shrink behavior of clay minerals). MPR can also be used to combine multiple predictors
to obtain new TFs (e.g., soil and land use predictors for plant root parameters, or topology and climate for snow parameters).



Most importantly, MPR enables users to specifically consider multiple commonly neglected uncertainty sources inherent in the geo-physical data, TF, and upscaling function. It is valuable for large-scale environmental models, where there is a current lack in effective parameter estimation, sensitivity analysis, and calibration (Beck et al., 2016).

*Code availability.* The software is publicly available at git.ufz.de/CHS/MPR and uses git for version control. The current version is 1.0.0 and the code is referenced by a Zenodo ID at https://zenodo.org/record/4650513. The code is published under the GNU GPLv3 license.

The code can be compiled by any recent Fortran compiler supporting the Fortran2008 standard and needs the `netcdf-fortran` library. In order to automatically add TFs to the code with a preprocessing script, the Python library f90nml must be installed.

The documentation framework FORD is used to create a https://chs.pages.ufz.de/MPR/index.html, which hosts a tutorial,
documentation, and extensive overview of the source code.

## Appendix A: Data processing

### A1  Processing of input data for minimal working example

1. Download the data from the server https://files.isric.org/soilgrids/data/recent/. The bulk density (fine earth), clay content, sand content, and soil organic carbon content (fine earth fraction) for all available soil depths at the original resolution
of 250m.

2. Transform the data from the tiff to the NetCDF format, clip the selected domain and merge the different layers into one file per variable (script in annex).

3. Download the target grids (e.g. for the ICON model (Zängl et al., 2015) grids, refer to http://icon-downloads.mpimet. mpg.de/mpim_grids.xml) following the SCRIP convention (Jones, 2010) for storing grids in the NetCDF format.

4. Select an example TF from the literature (e.g. Weynants et al. (2009)).

5. We constructed a configuration file *mpr.nml* for MPR in the native Fortran namelist format (Fig. 2).

## Appendix B: Model descriptions

### B1  mHM

mHM conceptualizes the dominant hydrological processes on the land surface through multiple reservoirs. The processes
of canopy interception, snow accumulation and melting, water infiltration into the soil and percolation to the groundwater, evaporation and transpiration, runoff generation, and river routing are accounted for on a spatially explicit grid. The model has been applied in a wide range of applications and has been shown to be able to fulfill the flux-matching criterion over multiple scales (Samaniego et al., 2017).





### B2   SoilHydroGrids

An increasing number of publications on high-resolution land surface datasets has led to the development of derivative datasets providing model parameters. Usually these datasets are available for a fixed resolution and domain only. Here, we demonstrate how MPR can be used to apply the TFs and remap the result on the domain and resolution as required. MPR is capable of reproducing the EU-SoilHydroGrids dataset (SHG) (Tóth et al., 2017) at a given 250m and 1km resolution.

### B3   Noah-MP

Noah-MP simulates the terrestrial water, energy, and carbon budget and estimates fluxes between various storage components in the biosphere, lithosphere, and hydrosphere. Its predecessor Noah (Ek et al., 2003) was superseded by Noah-MP by implementing multiparameterization options and improved physics for various ecohydrological processes. For each grid cell, the vertical model structure was discretized into one canopy layer using a semi-tile approach, three snow layers, four soil layers, and an unconfined subsurface layer.

### B4   HTESSEL


It calculates water, energy, and carbon fluxes and storage across the land surface. HTESSEL uses a tiling approach to represent different land covers within one model grid cell. It uses 20 plant functional types to describe vegetation and constant soil properties throughout the soil column. The soil has a standard depth up to 2.89 m.

### Appendix C:  Model configuration for application

### C1   mHM


MPR is required in order to reproduce the model parameters created by the internal version of MPR in mHM version 5.10 (Samaniego et al., 2019), and consequently, the same model results. The fact that mHM incorporates various autocalibration approaches which need control over the parameter estimation process necessitates a tight coupling of mHM to MPR in Fortran. We configured MPR to represent the complex interplay of model parameters. The configuration for mHM

encompassed the use of over 100 different Data_Array instances with over 60 TFs (see configuration in Supplement). The mHM code was refactored and adapted to allow for the passing of global parameters from mHM to MPR, and to allow the effective parameter fields to be received.

### C2   SoilHydroGrids

It is based on the SG dataset (Hengl et al., 2017) at 250m and an aggregated 1km resolution. Linear functions as well as

decision tree-based functions were used for the TFs (Tóth et al., 2015). A collection of relevant soil hydraulic parameters for the European domain are provided in SHG. We selected the subdomain of the Netherlands for the parameter saturated hydraulic conductivity, as it shows a large degree of variability in this region. The MPR configuration file is attached in the Supplement.





## C3    Noah-MP

We use the default WRF-Hydro parameterization process (version 3.6), at NCAR (2020), except for the radiative transfer op-
tion where a modified two-stream option was used. Meteorological model forcings were taken from the NLDAS-2 dataset (Xia
and NCEP/EMC, 2009). The $1/8°$ spatial resolution and hourly temporal resolution of the forcing variables (air temperature,
precipitation, specific humidity, wind speed, surface pressure, downward shortwave radiation, and downward longwave radia-
tion) also constitute the chosen model resolution for Noah-MP. The SG dataset (Hengl et al., 2017) was used to estimate soil
parameters, whereas the MODIS-IGBP (FRIEDL et al., 2010) dataset was used to derive vegetation parameters. In the default
setup, soil textural data was averaged over the soil column and classified into 12 classes using the Staff (1993) scheme. Finally,
the dominant type within a model grid cell was used. The exact parameters were then derived from class-specific default values
provided in the lookup tables. These default values were derived by applying a set of TFs (Cosby et al., 1984) to the mean
textural properties of the respective soil class. Vegetation parameters were also estimated based on the default values for each
effective land cover class. The Noah-MP model version 3.9 was used and slightly modified to explicitly allow for two specific
spatially distributed model parameters to be read. We used the default soil layering of horizon boundaries at depths of 0.1, 0.3,
0.6, and 1 m.

In addition to this default setup, we used MPR to estimate the soil parameters SATDK (soil saturated hydraulic conductivity)
and MAXSMC (maximum soil moisture content). Noah-MP shows a substantial sensitivity to both of these parameters along
a gradient of hydro-climate conditions in CONUS (Cuntz et al., 2016). The two parameters were estimated directly on a high-
resolution soil dataset using the following continuous TFs: TF 1 (Cosby et al., 1984), TF 2 (Saxton and Rawls, 2006), and
TF 3 (Vereecken et al., 1989, 1990). TF 1 is used in Noah-MP as a default option, TF 2 was later introduced as an option in
Noah-MP version 4.0, and TF 3 was chosen in this study to demonstrate the effect of a TF based on soil samples from outside
the study domain. The arithmetic mean was used to upscale the parameters to the model resolution, except for the vertical
scaling of $K_s$ along the soil horizons for which the harmonic mean was used. The parameters REFSMC (soil moisture content
at field capacity) and WLTSMC (soil moisture content at wilting point) were rescaled by the ratio of the default and modified
$\theta_s$ values ($\theta'_{ref} = \theta_{ref} * \theta'_s/\theta_s$).

The Noah-MP model was run for 28 years at an hourly time step from 1980 to 2007. We allowed the model to run for an
entire period and used the resulting state variables as initial conditions for the final run. The final evaluation period covered the
decade 1991–2000. The hourly simulation results were aggregated to mean annual values.

## 635   C4    HTESSEL

Meteorological forcing data were taken from the ERA5 dataset (ECMWF, 2019). The $1/4°$ spatial resolution and 3-hourly
temporal resolution of the forcing variables (air temperature, precipitation (rain and snow), specific humidity, wind speed, sur-
face pressure, downward shortwave radiation, and downward longwave radiation) also constitute the chosen model resolution
for HTESSEL.





```fortran
1    abstract interface
       function transfer_func_alias(x, param) result(func_result)
         ! import the double precision kind specification and custom
             type
         import dp, InputFieldContainer
         !> an array containing the predictor variables (access values
             through `data_p` property)
6        type(InputFieldContainer), intent(in) :: x(:)
         !> an array containing the TF parameters
         real(dp), intent(in) :: param(:)
         !> the resulting TF result
         real(dp), allocatable :: func_result(:)

11       ! ! allocate the func_result to the size of the predictors
             (all have the same size)
         ! allocate(func_result(size(x(1)%data_p)))
         ! ! enter the TF function here
         ! func_result = x(1)%data_p + x(2)%data_p * param(1)

16       end function transfer_func_alias
     end interface
```

**Figure D1.** Template for user-defined TFs as an abstract interface.

We used a process parameterization based on the default configuration presented in the development branch CY47R1 of HTESSEL (nemk_CY47R1.0_v6b_cmflood_mpr). In addition to this default setup, we used MPR to estimate the six soil parameters of the Mualem-van Genuchten model for the hydraulic conductivity curve (Equation 1) and soil moisture retention curve (Equation 2). These parameters were estimated directly on the SG dataset (Hengl et al., 2017) using the following TFs: categorical TF 1 based on lookup-table values (HYPRES, 1997), continuous TF 2 (Zacharias and Wessolek, 2007) with

the estimation of only the four parameters of Equation 2, and continuous TF 3 (Wösten et al., 2001). Soil textural data was averaged over the soil column for TF 1, classified into 7 classes using the HYPRES soil texture triangle (HYPRES, 1997; Wösten et al., 1999), with some additions for organic soils. Finally, the dominant type within a model grid cell was used. The exact parameters were then derived from class-specific default values provided in the lookup tables. TF 1 is used in HTESSEL as a default option. The arithmetic mean was used to upscale the parameters to the model resolution, except for the vertical

scaling of $K_s$ along the soil horizons, for which the harmonic mean was used.

The HTESSEL model version CY47R1 was used and modified to explicitly allow for the reading of spatially distributed model parameters. We used the default soil layering of horizon boundaries at depths of 0.07 m, 0.28 m, 0.1 m, and 2 m. Due to limitations in the HTESSEL solver for the soil physics processes, we enforced vertically homogeneous soil parameters.

The HTESSEL model was run for 8 years at a daily time step from 1979 to 1986. We allowed the model to run for an entire

period and used the resulting state variables as initial conditions for the final run. The final evaluation period covered the years 1979 to 1986. The 3-hourly simulation results were aggregated to mean annual values.

## Appendix D: Templates for user-defined functions





```fortran
 1    abstract interface
        real(dp) function upscale_func_alias(array, weights, p)
          ! import the double precision kind specification
          import dp
          !> the array of subgrid values (no missing values)
 6        real(dp), dimension(:), intent(in) :: array
          !> the array of weights (same shape as array)
          real(dp), dimension(:), intent(in), optional :: weights
          !> an optional parameter passed to the function (power mean)
          real(dp), intent(in), optional :: p
11      end function upscale_func_alias
      end interface
```

**Figure D2.** Template for user-defined upscale operators as an abstract interface.





**Appendix E: Inventory of operators in transfer functions and for upscaling**





**Table E1.** Table of operators that can be specified in a string for the property `transfer_func` in the configuration file *mpr.nml*. They are directly parsed to the Fortran code.

| operator | type | description |
| --- | --- | --- |
| + | binary operator | - |
| − | binary operator | - |
| ** | binary operator | - |
| * | binary operator | - |
| / | binary operator | - |
| max | binary operator | - |
| min | binary operator | - |
| exp | unary operator | - |
| sqrt | unary operator | - |
| log10 | unary operator | - |
| log | unary operator | - |
| abs | unary operator | - |
| ( | delimiter | - |
| ) | delimiter | - |
| if | conditional statement | - |
| else | conditional statement | - |
| then | conditional statement | - |
| where | conditional statement | - |
| end | conditional statement | - |
| <= | relation operator | - |
| < | relation operator | - |
| >= | relation operator | - |
| > | relation operator | - |
| == | relation operator | - |
| .and. | logic notation | - |
| .or. | logic notation | - |
| .not. | logic notation | - |
| sin | trigonometric function | - |
| cos | trigonometric function | - |
| tan | trigonometric function | - |
| tanh | trigonometric function | - |
| acos | trigonometric function | - |
| asin | trigonometric function | - |
| atan | trigonometric function | - |
| atan2 | trigonometric function | - |
| cosh | trigonometric function | - |
| sinh | trigonometric function | - |





**Table E2.** Table of operators that can be specified in a string for the property `upscale_ops` in the configuration file *mpr.nml*. The values $x$ of the array with size $n$ (with indices $i$) are passed to the operator with weights $w$ of the same size.

| operator | equation | description |
|---|---|---|
| $p, p = 0$ | $\prod_{i=1}^{n} x_i^{w_i}$ | geometric mean |
| $p, p \in \mathbb{R}$ | $\left(\sum_{i=1}^{n} w_i x_i^p\right)^{\frac{1}{p}}$ | power mean |
| `min` | $\min\{x_i\}$ | minimum |
| `max` | $\max\{x_i\}$ | maximum |
| `sum` | $\sum_{i=1}^{n} x_i$ | sum |
| `var` | $\sum_{i=1}^{n} w_i (x_i - \overline{x})^2$ | variance |
| `std` | $\sqrt{\sum_{i=1}^{n} w_i (x_i - \overline{x})^2}$ | standard deviation |
| `laf` | $\min\{x_j \mid j \in 1, ..., \lvert x \rvert : \forall i : w_i <= w_j\}$ | largest area fraction |





*Author contributions.* RS and ST designed the study. RS was the primary code developer with support from ST. RS conducted all simulations
and evaluations of the application, except for HTESSEL (simulations conducted by MK ). RS, ST, RK, SA, and LS prepared the manuscript.

*Competing interests.* The authors declare that they have no conflict of interest.

*Acknowledgements.* We thank the developers of the scientific softwares that enabled this study, namely: mHM, Noah-MP, and HTESSEL
developers, many NumFOCUS-sponsored Python libraries (above all numpy, xarray) and many privately-run projects (f90nml, FORD, tikz,
dot2tex). We are grateful to Christel Prudhomme and Florian Pappenberger from ECMWF for making the HTESSEL model available to us
and the group from Gianpaolo Balsamo for providing extraordinary support with the HTESSEL Fortran code. This project contributes to the
Helmholtz ESM project https://www.esm-project.net/.





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
