# Peer review of "MPR 1.0: A stand-alone Multiscale Parameter Regionalization Tool for Improved Parameter Estimation of Land Surface Models"

_Geoscientific Model Development, 2021_

## Author Response (AR1)

**Response to Reviewer 1**

We would like to thank the reviewer for her/his time and effort in revising our manuscript and the provided comments. We address her/his comments below. Reviewer comments are *italic*.

*This paper describes the stand-alone version of MPR and its adaptation for two different land surface models. MPR is a well-established regionalization technique to estimate spatially explicit model parameters, and the availability of a stand-alone version could potentially contribute to significant model improvement in some land surface models and hydrological models. For modelers who already plan on incorporating MPR, this paper would provide much needed guidance. However, if the purpose of this paper is to convince other large-scale environmental modelers to adapt MPR (which would surely merit publication in GMD), significant improvement in paper organization and results presentation is needed.*

**Paper Organization**

*Some parts of the paper read more like a technical document than a scientific paper. For example, a lot of material in sections 3.1-3.3 is better suited for supplementary materials.*

We understand the motivation of the referee that we want to expand our knowledge about the environment by conducting science. We share this motivation and often use numerical models as tools. We are deeply frustrated about the lack of documentations of numerical models in the past that limit the reproducibility of scientific experiments. For this reason, we decided to make the effort to document this MPR tool and make it available to interested scientists. We did submit this manuscript as manuscript type "Model description papers". We followed the instructions provided by GMD journal (https://www.geoscientific-model-development.net/about/manuscript_types.html#item1, access: Sep 22nd 2021) to create this manuscript. The purpose of the manuscript is as stated on the web page: "Model description papers are comprehensive descriptions of numerical models which fall within the scope of GMD. The papers should be detailed, complete, rigorous, and accessible to a wide community of geoscientists. In addition to complete models, this type of paper may also describe model components and modules, as well as frameworks and utility tools used to build practical modelling systems, such as coupling frameworks or other software toolboxes with a geoscientific application. The GMD definition of a numerical model is generous, including statistical models, models derived from data (whether model output or observational data), spreadsheet-based models, box models, 1-dimensional models, through to multi-dimension mechanistic models."

This is why our manuscript does not include a research question or research hypothesis which is a main requirement for a scientific paper. For this reason, we created these detailed description in Section 3 that we think is necessary to fully understand the inner workings of MPR.

*Instead of these nitty gritty details, the readers may prefer to see things such as the concept of MPR explained (like how to apply TF), computing time required at a representative configuration (hardware and spatial resolution), why the experiments have to be performed online, and the pros and cons of the MPR method.*

The concept of MPR is explained in the introduction (line 48ff.) and conceptual papers by Samaniego et al., 2010 and Kumar et al., 2013, that we referenced at the beginning of the paragraph. How to apply TFs is explained in detail in section 3.4.4. Following your suggestion, we will add hardware requirements, run time, and spatial resolution for the examples shown in the manuscript (lines 182ff.). We do not state that experiments need to be performed online, especially as MPR needs to be regarded as pre-processor that can run in both offline and online simulations. Following your suggestion, we will explicitly add advantages and disadvantages of the MPR concept and tool in section 2.3 (lines 175ff.). These are the advantages and disadvantages that we plan to include are:

Advantages:

- use of high-resolution data sets for parameter estimation
- calculating model parameters at the highest resolution possible before aggregating to a model resolution
- flexibility to estimate parameters using arbitrary number of predictor variables, Transfer Functions and Upscaling Operators
- low run times because of Fortran implementation
- increased reproducibility of numerical work flows

Disadvantages:

- high memory requirement due to calculation at high resolution
- software dependency on Fortran compiler and netcdf library
- netcdf is the only format supported for input and output
- model coupling works only if (distributed) parameters are accepted (in netcdf format)

*In Section 3.4: Please also explain the difference with old MPR release. What issues prevent the old version from being used by other models (I assume other models can also use the parameter files)? Which improvements were made to solve these issues?*

We will expand Section 3.4 (lines 352ff.) on differences between the new and old version repeating some of the statements made in the introduction (l. 57ff). The old version is hard-coded in the source code of the mesoscale Hydrologic Model (mHM). The user cannot change the TFs and upscaling operators for another purpose. Additionally, the obtained parameter fields are stored in a format internal to mHM that cannot be used for other models easily.

Results Presentation

*For other LSM modelers, Figure 5 would be most interesting. One would naturally ask whether the use of MPR result in better ET, which transfer function (TF) was used (and how to choose appropriate TF), and whether the authors have noticed some similarities (i.e., if improved results are shown in both cases) in these two LSMs. These issues are important for new users to judge the applicability of MPR. I would recommend the authors to consider using some open-source benchmarking tools such as ILAMB to comprehensively assess whether the simulation results of the two LSMs have improved in various key variables.*

We would like to thank you for this comment. The purpose of this Figure is to demonstrate that using different TFs has an impact on simulated fluxes. This shows that the MPR method is relevant. The question which TFs to choose to minimize a given objective function is out of the scope of this paper. However, we are working in this direction and did find large improvements in streamflow simulations when optimizing soil parameters of Noah-MP and HTessel, but cannot show results here.

*Figure 3: The examples presented in this paper discussed theta_s and K_s, while LAI was not discussed. So why not show dependency graphs for these two parameters instead?*

In Figure 3, LAI was chosen as it nicely highlights the use of multiple dimensions and uses simplistic TFs. The derivation of the K_s and theta_s parameters in mHM is rather complex as a correction for organic matter in tillage layers is applied. Thus, we refrained from using this as an example in the manuscript. Please see the dependency graph for these two soil parameters attached (Figure 1).

**Some Minor Points**

*L372-376: Have you contacted that paper's authors for this issue?*

L372-376 No, we have not contacted the authors.

*L429-L430: How?*

The example refers to Noah-MP. The model comes with a default parameter table. Likely, this is derived from literature or previous versions and is assumed to work for a given set of input data (soil, landuse, etc.). We assume, that the model bias in key variables observed by Ma et al. (2017) stems in part from the parameters used. Applying MPR will alter the parameter distributions so that bias might be minimized by model parameter calibration. We will add the following sentence at the end of the paragraph (line 451ff): "For example, MPR and Noah-MP can be executed subsequently by an optimization algorithm. The optimizers draws new parameter sets for MPR that result in updated soil parameter maps for Noah-MP. In turn, updated ET fields are calculated by Noah-MP."

[Figure]

Figure 1: Dependency graph for soil parameters K_s and theta_s.

*L514: "optimization applications is low" do you mean "optimization applications is high"?*

L514 This became obsolete by implementing the proposed changes from RC3.

*Code availability: please provide sample data and installation guild if possible. Data to reproduce the figures cannot be downloaded without contacting UFZ.*

Sample data are already part of the code base (in tests section). Download instructions will be provided. We removed the barrier for the access to the manuscript's meta information so the reviewer does not need to reveal her/his identity while downloading.

**References**

Ma, N., Niu, G.-Y., Xia, Y., Cai, X., Zhang, Y., Ma, Y., and Fang, Y.: "A Systematic Evaluation of Noah-MP in Simulating Land-Atmosphere Energy, Water, and Carbon Exchanges Over the Continental United States", Journal of Geophysical Research: Atmospheres, 122, 12,245– 12,268, https://doi.org/10.1002/2017JD027597, https://agupubs.onlinelibrary.wiley.com/doi/abs/10.1002/2017JD027597, 2017.

Samaniego, L., Kumar, R., and Attinger, S.: Multiscale parameter regionalization of a grid-based hydrologic model at the mesoscale, Water Resources Research, 46, https://doi.org/10.1029/2008WR007327, https://agupubs.onlinelibrary.wiley.com/doi/abs/10. 1029/2008WR007327, 2010

Kumar, R., Samaniego, L., and Attinger, S.: Implications of distributed hydrologic model parameterization on water fluxes at multiple scales and locations, Water Resources Research, 49, 360–379, https://doi.org/10.1029/2012WR012195, https://agupubs.onlinelibrary.wiley.com/ doi/abs/10.1029/2012WR012195, 2013b

**Response to Reviewer #2**

We would like to thank the reviewer for her/his time and effort in revising our manuscript and the provided comments. We address her/his comments below. Reviewer comments are *italic*.

**General comment**

*This paper addresses a key topic for earth system modelling related to the physical parametrization and the associated uncertainties. For land surface models the representation of soil water transfer relies on transfer functions which relate soil texture properties (that can be accessed from soil map at global and regional scale) to the soil hydraulic parameters. As largely documented by previous studies, these transfer functions are important sources of uncertainties which can be related to their calibration and/or their application at coarse grid resolution. This paper presents a new configuration of the Multiscale Parameter Regionalization framework used for soil water parametrization in land surface model and shows its ability and flexibility to generate continental-scale soil hydraulic parameters at distinct scales. The tool is evaluated for the simulation of evapotranspiration flux using two land surface models largely used by the land surface community.*

*My main concern for this paper, is that most of the content is dedicated to the development of the tool which include a large part of technical and programming components. While this information is of importance for the user community, the scientific part of the paper dedicated to the evaluation of the tool is quite limited. I wonder whether this imbalance between the technical development and the actual scientific outcomes can be a problem for publication in GMD. My main recommendation is major revision to emphasize the scientific value-added of the presented parametrization tool. For this reason, I provide very light feedbacks because I think that the content needs to be modified before doing any additional review.*

We use hydrologic models and earth system models to improve our understanding about the earth system. Within our research, MPR is a tool that we want to expand to other models and make available to other scientists. To do so, we need a stand-alone version of MPR in which we can easily change the TFs and upscaling operators. We describe this newly developed stand-alone version in this model description paper. This is the reason why there is a comprehensive technical description in this manuscript. Additionally, we demonstrate how MPR can be applied and that the application of MPR leads to substantial changes in flux estimates. This is the content of Section 4. In this Section, we show that the new MPR implementation gives the same results as the previous one for the mesoscale Hydrologic Model (mHM). We apply MPR to reproduce soil hydraulic parameters and show that MPR allows to easily obtain results at different configurations. Finally, we apply MPR to calculate parameters for two state-of-the-art LSMs and quantify the impact on long-term fluxes. In summary, this highlights the relevance of MPR. The scientific requirement for

MPR has also been exhaustively discussed before with great detail in Samaniego et al. (2017). While interesting scientific questions arise from the presented material, like how to choose TFs for a given parameter and objective, what is the impact of predictor variables on obtained model parameters etc., it has to be clear that there are no scientific questions within this manuscript. MPR, however, allows to ask and answer scientific questions in a rigorous, reproducible way which is the scientific value-added of the presented MPR software. In line with comments from reviewer 1, we will add advantages of MPR in Section 2.3 that should further highlight the scientific value-added (lines 175ff.).

**Specific comments**

*Section 2: This section is quite atypical for a scientific paper. Part of the content concerning the rationale/motivation of this study (e.g. section 2.3), background information on parameter estimation workflow (section 2.1 and 2.2) should be moved to a dedicated paragraph in Introduction. There are also methodological statements that belong to a method section. The IT details given in 2.2 (e.g. line 148-152) may not be relevant in the main part of the paper, I suggest including them in additional material.*

We agree with the reviewer that this is an atypical section for a manuscript. We introduced this section to provide an explicit example that is easy to understand for modellers, a group we would like to address explicitly. Following the advice from reviewer 3, we will move Section 2.4 to the beginning of Section 3 (line 191ff). Following your advice, we will shorten lines 148-152 (now lines 138ff.).

*The current structure of the paper is difficult to read. The data section which is in appendices should be a main section of the paper. Also, a method section is clearly missing. Overall the structure of the paper needs to be improved since sometimes method, model description and results are included in the same section that does not facilitate the reading of the paper.*

The data section provides information on the data fields used as predictor variables. They are deliberately moved to the appendix. We believe the information is not essential for understanding the functionality of the MPR tool and it distracts users from the core messages of the results section. The main purpose of the manuscript is the introduction of the MPR tool, not the benchmark of the land-surface models, which are chosen as example. This paper introduces a new software tool. Thus, section 3 is the method section.

*The objective of section 4.4.1 is not clear, why comparing the results from both models ? as underlined the models have very different configurations that make their comparison quite hazardous. Maybe I missed the objective here.*

We would like to thank you for this comment. We will elaborate more the objective of section 4.4.1 that will be added to the introduction. The objective is to show the use of MPR for model intercomparison studies. MPR enables model comparison studies specifically comparing the use of predictor variables,

transfer functions and upscaling techniques in each model. Yet, comparisons can be hazardous, as model parameterizations and forcings can be different. This is the case here, as pointed out by the reviewer. We will explicitly add in the beginning of the Section (lines 493ff.) that it is not the purpose of this Section to provide a rigorous model intercomparison. However, we show that for the examples of Noah-MP and HTessel, conclusions of the impact of model parameters on long-term fluxes can be made for Noah-MP, but not for HTessel (due its more complex process implementations). Since both models are run for the same domain, we find it essential to discuss the differences between them.

**References**

Samaniego, L., Kumar, R., Thober, S., Rakovec, O., Zink, M., Wanders, N., Eisner, S., Müller Schmied, H., Sutanudjaja, E. H., Warrach- Sagi, K., and Attinger, S.: Toward seamless hydrologic predictions across spatial scales, Hydrol. Earth Syst. Sci., 21, 4323–4346, https://doi.org/10.5194/hess-21-4323-2017, https://www.hydrol-earth-syst-sci.net/21/4323/2017/, 2017

**Response to Reviewer #3**

We would like to thank the reviewer for her/his time and effort in revising our manuscript and the provided comments. We address her/his comments below. Reviewer comments are *italic*.

**Brief summary of the paper**

*The paper presents descriptions of stand-alone tool that aims to assist hydrologic modelers with parameter regionalization through Mulit-scale Parameter Regionalization method (Samaniego et al., 2010). The paper also provides examples of distributed soil parameters computed with this new tool and applied a few hydrologic and land surface models and their impacts on the water flux (ET) simulations.*

**Overall comments**

*First of all, I recognize the great potential of MPR approach for distributed parameter estimation for various environmental models, and because MPR was imbedded into mHM system originally, it is a good to make MPR routine stand-alone routine and make them generic tool so that MPR can be used for the other models. The paper's motivation is clear.*

We appreciate that the reviewer fully understands our motivation to write this model description manuscript.

*That being said, real challenge in MPR approach is 1) how to determine useful geophysical predictors, 2) how to determine a transfer function for each model parameter and 3) how to determine appropriate scaling operators so that fluxes from various spatial resolutions match over the larger areas. I wished to see how this stand-alone tool could help to tackle those challenges (particularly challenge 2 and 3), and wonder if the authors could discuss more about these. I did see brief statements in line 87-88, page 3– "Emerging methods for the development of TFs do exist (Klotz et al., 2017; Feigl et al., 2020; Merz et al.,2020). MPR provides the interface to link these tools to distributed environmental models". I am really curious how this tool can interface with the tool that optimize transfer function form (if possible).*

The reviewer has an excellent understanding of the MPR method and we share the same curiousity. We want to address the questions outlined by the reviewer and to do so, we need a flexible and easy to use implementation of MPR. This implementation is presented in the current manuscript. It goes beyond the scope of the manuscript to present how this implementation of MPR can be used in a parameter and transfer function optimization. We are working in this direction and will expand the section outlined by the reviewer (lines 251ff.). First preliminary results can be found in Feigl et al. (2021, only in german).

*After I read the paper, I felt the paper is about software design leaving out important MPR concept, probably due to the length of section 3 where lots of*

*technical descriptions of the code are described. Therefore, I would suggest focus more on the capability of this tool for parameter estimation. I think this is the biggest issue for this manuscript.*

We would like to thank the author for this comment. As this is a model description manuscript (we submit this manuscript as model description type to GMD), we followed the advice provided by GMD on this type of manuscript (https://www.geoscientific-model-development.net/about/manuscript_types.html#item1, access: Sep 22nd 2021). To our understanding, we provide the very detailed description of the MPR implementation to follow the requirements for this paper type. The concept is introduced in the introduction (l. 48ff.). This paragraph also contains the references to Samaniego et al. (2010, WRR) and Kumar et al. (2013, WRR), which provide a detailed introduction of the MPR concept for readers who are interested.

*In summary, I think the author did excellent job developing tool, but the paper itself requires major revisions so hydrologic modelers (who are not likely to be software engineers) could follow better. I would provide some more comments bellow.*

**Specific comments**

*I think introduction is overall good.*

We would like to thank the reviewer for this comment.

*Most descriptions in the section 2 are appropriate except for section 2.4. I am not sure if section 2.4 is really important— especially some lengthy discussions on netcdf (line 188 – 195). Also, some shorter version of the second paragraph (L197 – 205) can be moved to section 3?*

Following this comment and the advice from reviewer 2, we moved Section 2.4 into section 3 (line 191ff.).

*Section 3.2. I think it is important to show the parameter estimation needs to consider parameter dependencies, but an example provided here may be over-complex due to lengthy descriptions of requirement of broadcasting array if coordinates in two data array do not match (I am not sure if the user needs to understand this or the tool does this behind scene). My suggestion would be to emphasize parameter dependencies (could be one soil parameter example; e.g., baseflow coefficient may be related hydraulic conductivity, may be helpful) and the tool can provide flexibility to define parameter dependencies.*

We will explicitly mention parameter dependencies in Section 3.3 and highlight that MPR allows to account for parameter dependencies. Please see also Figure 1 showing the parameter dependencies.

*Section 3.3. I think this is where I have trouble in this paper. This section focuses on the software design. I am not sure if readers (most likely hydrologic modelers) need to understand detailed information on how software works at the*

[Figure]

Figure 1: Dependency graph for soil parameters K_s and theta_s.

*code level (e.g., connections between objects). Section 3.3.4 and section 3.3.5 are core part of MPI, so I would think I would replace the current descriptions with the flexibility and capability of the tool to form transfer function and scaling operators.*

We thank the reviewer for this comment. We expect two groups of interested readers. First, model developers for which it is important to know that MPR can be used as a library. For this group, the overview of the different objects in MPR is required. Second, hydrologic modelers for which these details might be not needed. For this group, we will improve the readability by making more references in the text to either Fig. 1 or Fig. 3, which provide an example work flow of MPR, to provide more guidance to the reviewer through these Sections (lines 251ff., 287ff., 298, 320ff., 337ff.).

*Section 4.5. It is important to discuss computational cost associated with use of high-resolution geophysical data over a large domain (e.g., continent or global), but other discussions appear to be unnecessary.*

As suggested by the reviewer, we will shorten this paragraph by removing all bullet points but the first one and we will remove the paragraph on compiler compatibility (lines 532ff.).

**References**

Feigl, M., Herrnegger, M., Schweppe, R. et al. Regionalisierung hydrologischer Modelle mit Function Space Optimization. Österr Wasser- und Abfallw 73, 281–294 (2021). https://doi.org/10.1007/s00506-021-00766-0

Samaniego, L., Kumar, R., and Attinger, S.: Multiscale parameter regionalization of a grid-based hydrologic model at the mesoscale, Water Resources Research, 46, https://doi.org/10.1029/2008WR007327, https://agupubs.onlinelibrary.wiley.com/doi/abs/10.1029/2008WR007327, 2010

Kumar, R., Samaniego, L., and Attinger, S.: Implications of distributed hydrologic model parameterization on water fluxes at multiple scales and locations, Water Resources Research, 49, 360–379, https://doi.org/10.1029/2012WR012195, https://agupubs.onlinelibrary.wiley.com/doi/abs/10.1029/2012WR012195, 2013b

---

## Author Response (AR2)

**Response to Editor**

Dear Carlos Sierra,

we would like to thank you for your positive assessment of our manuscript and your detailed recommendation for further improvements. Following your advise, we added a road map to the manuscript at the end of the introduction (l. 90 to 104). We provide a short summary of the different Sections and added recommendations on which are worth reading for the different audiences we are targeting. We are optimistic that these modifications satisfy your requirements.

We would like to thank you for your time and effort handling our manuscript.

Best regards,

Stephan Thober

---

## Author Response (AR3)

**Response to Editor**

Dear Carlos Sierra,

we have removed the copyright statement on page 1.

Best regards,

Stephan Thober